# MambaSCI: Efficient Mamba-UNet for Quad-Bayer Patterned Video Snapshot Compressive Imaging

**Zhenghao Pan**[1,*], **Haijin Zeng**[2,*], **Jiezhang Cao**[3], **Yongyong Chen**[1,†], **Kai Zhang**[4], **Yong Xu**[1]

[1] Harbin Institute of Technology (Shenzhen), [2] Ghent University,
[3] Harvard University, [4] Nanjing University

## Abstract

Color video snapshot compressive imaging (SCI) employs computational imaging techniques to capture multiple sequential video frames in a single Bayer-patterned measurement. With the increasing popularity of quad-Bayer pattern in mainstream smartphone cameras for capturing high-resolution videos, mobile photography has become more accessible to a wider audience. However, existing color video SCI reconstruction algorithms are designed based on the traditional Bayer pattern. When applied to videos captured by quad-Bayer cameras, these algorithms often result in color distortion and ineffective demosaicing, rendering them impractical for primary equipment. To address this challenge, we propose the MambaSCI method, which leverages the Mamba and UNet architectures for efficient reconstruction of quad-Bayer patterned color video SCI. To the best of our knowledge, our work presents the first algorithm for quad-Bayer patterned SCI reconstruction, and also the initial application of the Mamba model to this task. Specifically, we customize Residual-Mamba-Blocks, which residually connect the Spatial-Temporal Mamba (STMamba), Edge-Detail-Reconstruction (EDR) module, and Channel Attention (CA) module. Respectively, STMamba is used to model long-range spatial-temporal dependencies with linear complexity, EDR is for better edge-detail reconstruction, and CA is used to compensate for the missing channel information interaction in Mamba model. Experiments demonstrate that MambaSCI surpasses state-of-the-art methods with lower computational and memory costs. PyTorch style pseudo-code for the core modules is provided in the supplementary materials. Code is at `https://github.com/PAN083/MambaSCI`.

## 1 Introduction

In recent years, there has been significant progress in enhancing imaging quality in smartphone image sensors. One notable trend is the adoption of the quad-Bayer Color Filter Array (CFA) pattern [1]. Smartphones such as the iPhone 14 Pro/Max, vivo X90 Pro+, Xiaomi 13S Ultra, and OPPO Find X6 Pro utilize quad-Bayer array to enhance image quality in low-light conditions and offer higher resolution for mobile photography [2, 3]. Unlike the traditional RGGB Bayer CFA pattern, the quad-Bayer pattern expands each pixel into four sub-pixels and arranges them periodically [4, 5], as depicted in Fig. 1(a). This arrangement allows for larger pixels by

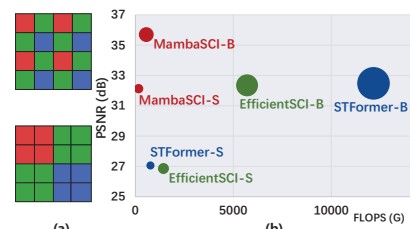

Figure 1: (a) Bayer CFA vs. Quad-Bayer CFA. (b) PSNR and FLOPS on color simulation videos (larger size means more parameters).

---

*Equal contribution. † Corresponding Author: Yongyong Chen (YongyongChen.cn@gmail.com)

38th Conference on Neural Information Processing Systems (NeurIPS 2024).

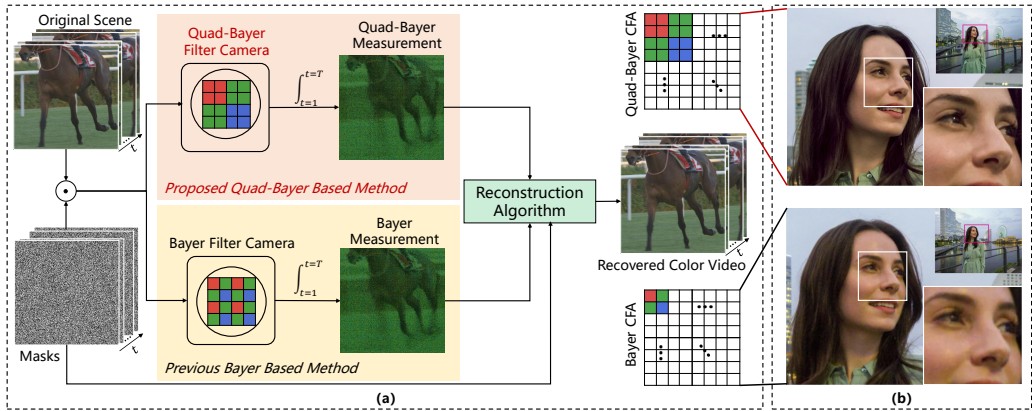

Figure 2: (a) Schematic diagram of the comparison between color video SCI based on the proposed quad-Bayer-based method and the previous Bayer-based method. (b) Photo taken by quad-Bayer CFA pattern (Sony IMX689) (top) and Bayer CFA pattern (bottom). One can see that the upper image is sharper with less noise.

combining neighboring pixels of the same color, resulting in more light intake compared to the Bayer pattern. Consequently, quad-Bayer sensors offer enhanced sensitivity and resolution for imaging tasks [6, 7, 8]. As illustrated in Fig. 2(b), quad-Bayer technology effectively mitigates resolution loss and enables the capture of low-noise photos in low-light environments. In summary, quad-Bayer sensors provide HDR capability [9] and improved color accuracy [10] while effectively mitigating resolution loss and capturing low-noise photos in low-light environments.

Color videos captured by traditional high-speed cameras incur high transmission and storage costs. To solve this issue, color video snapshot compressive imaging (SCI) [11], which comprises both hardware encoder and software decodes components, has been proposed. During the encoding phase, multiple raw video frames undergo modulation and compression using various masks to generate 2D measurements [12]. Subsequently, in the decoding stage, the desired high-speed color video is reconstructed from the acquired measurements and predefined masks.

So far, all color video SCI encoding and reconstruction algorithms have been designed based on the Bayer pattern [13, 14, 15, 16, 17]. As smartphone cameras continue to improve in pixel count and performance, the majority of videos are now captured using smartphones equipped with quad-Bayer patterns. However, our observation reveals that existing methods struggle to effectively reconstruct videos based on quad-Bayer patterns, often resulting in artifacts, color distortions, and incomplete demosaicing. This ineffectiveness poses a challenge for processing videos captured by smartphone cameras. Thus, in this paper, we aim to break through these challenges and introduce the first quad-Bayer-based color video SCI reconstruction method, as illustrated in Fig. 2(a).

When incorporating quad-Bayer in color video SCI reconstruction tasks, two challenges arise. One is ***how to efficiently manage quad-Bayer color video processing with reduced computational complexity.*** Current color video SCI reconstruction methods encompass model-based [13, 18, 19], iteration-based [14, 15], and End-to-End (E2E) approaches [20, 21]. However, model-based and iterative methods necessitate corresponding demosaicing models [6, 22, 23, 24], which have not been extensively explored for quad-Bayer fields regarding performance and model size trade-offs, leading to suboptimal reconstruction outcomes and inefficiencies. On the other hand, existing E2E methods, primarily transformer-based [17] and hybrid CNN-transformer [16] approaches, typically require significant parameters and computational resources, making it difficult to process long video sequences effectively. The high computational demands of transformers and the absence of a global attention mechanism in CNNs hinder their scalability to modern, lightweight architectures. Leveraging the advancements in State Space Models (SSMs) [25, 26, 27], modern SSMs like Mamba [28] have demonstrated the ability to effectively capture long-range dependencies while maintaining linear complexity relative to the input size. Furthermore, numerous experiments have illustrated that image-based Mamba [29, 30] achieves promising results and can match the performance of existing transformer [31, 32, 33] and CNN [34, 35, 36] models with smaller parameters. Therefore, we focus on exploring the multi-scale reconstruction capabilities of Mamba-UNet for quad-Bayer processing. This approach aims to create a lightweight design suitable for deployment on mobile devices. By leveraging the Mamba-UNet framework [37, 38, 39, 40], we can reduce parameters and FLOPS while still achieving state-of-the-art performance.

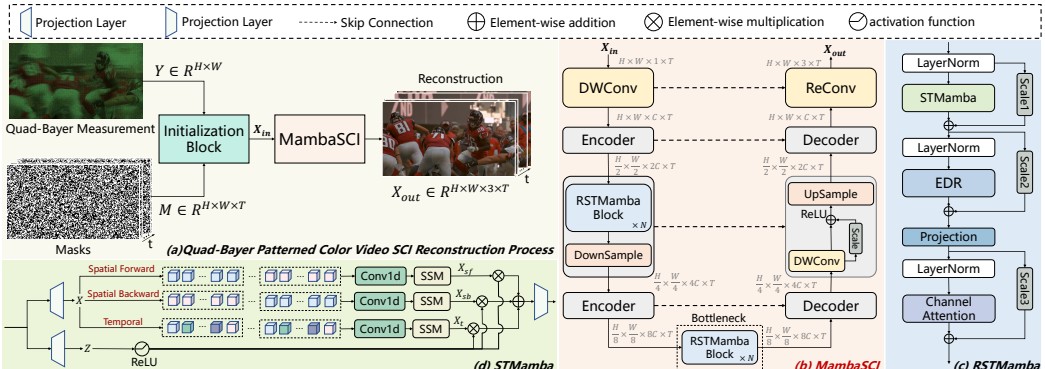

Figure 3: The proposed MambaSCI network architecture and overall process for color video reconstruction. (a) Quad-Bayer patterned color video SCI reconstruction process. It feeds quad-Bayer pattern measurement $\mathbf{Y}$ and masks $\mathbf{M}$ into the initialization block to get $\mathbf{X}_{in}$ and inputs it into MambaSCI network to get the reconstructed RGB color video $\mathbf{X}_{out}$. (b) The overall network architecture of the proposed MambaSCI network. (c) Structure of Residual-Mamba-Block (RSTMamba) with STMamba, EDR, and CA modules connected via residuals. The detailed design of EDR and CA is shown in Fig. 4. (d) STMamba. It captures spatial-temporal consistency via structured SSMs that enable parallel scanning in the spatial forward-backward and temporal dimensions.

The second challenge is ***how to eliminate motion artifacts to ensure clarity in dynamic video.*** Motion artifacts may arise from handheld camera instability and pixel merging in quad-Bayer patterns. To preserve scene clarity and edge details, we have customized the Residual-Mamba-Block, which integrates three key components: the Spatial-Temporal Mamba (STMamba), the Edge-Detail-Reconstruction module (EDR), and the Channel Attention module (CA) [41, 42]. The Residual-Mamba-Block further enhances long-range spatiotemporal dependence by leveraging residual connections and learning scales. STMamba replaces the self-attention module by performing parallel scanning of spatial and temporal dimensions, ensuring spatiotemporal consistency and enabling visual reconstruction of videos in a more lightweight manner. The EDR module enhances boundary sharpness perception and restores edge details lost due to compression and motion artifacts. By combining linear transformation with DWConv features, it integrates both local and global information. This approach strengthens the global features extracted by STMamba, fusing them with local details to more effectively capture complex edge structures. As a result, the module enables high-quality video reconstruction with improved edge clarity. The CA module compensates for the overlooked channel interactions in Mamba by weighting features based on channel importance, reducing the impact of artifact noise on reconstruction and thus improving the clarity of reconstruction results.

Building upon the foundation modules discussed earlier, we introduce a novel model called MambaSCI, which serves as the reconstruction algorithm for our proposed quad-Bayer-based SCI method as illustrated in Fig. 2(a). MambaSCI adopts a non-symmetric U-shaped encoder-decoder architecture with skip connections to enhance model efficiency. To incorporate spatial-temporal consistency at multiple scales, we introduce Residual-Mamba-Blocks at each encoding stage. Furthermore, we employ residual convolution in the decoding stage to reduce both parameters and computational complexity. The effectiveness of the MambaSCI model is demonstrated in Fig. 1(b), where it achieves superior performance compared to the SOTA methods while requiring fewer parameters and FLOPS. Pseudo-code detailing the core modules is provided in the supplementary materials.

In short, our contributions can be summarized as follows:

**(i)** We are the **first** to introduce Quad-Bayer CFA pattern into color video SCI to accommodate that most videos are captured by mobile photographers using quad-Bayer patterned smartphone cameras.

**(ii)** We are the **first** to use Mamba model for video SCI reconstruction. By integrating Mamba with a non-symmetric UNet, we employ a hierarchical encoder to capture spatial-temporal correlations at various scales, thus accelerating the model and enhancing reconstruction quality.

**(iii)** We customize the Residual-Mamba-Block, integrating STMamba, EDR, and CA modules with residual connections and learnable scales to enhance reconstruction quality and edge details.

**(iv)** MambaSCI outperforms other SOTA methods, requiring fewer parameters and FLOPS, and delivers superior visual results on 6 simulation color videos and 4 large-scale simulation color videos.

## 2 Related Work

### 2.1 Mathematical Model for Color Video SCI

For color video SCI systems, the original $T$-frame input video $\mathbf{X} \in \mathbb{R}^{H \times W \times 3 \times T}$ is given, along with the mask $\mathbf{M} \in \mathbb{R}^{H \times W \times T}$, where $H, W$ and $T$ denote color video's height, width and number of frames, respectively. As in previous Bayer-based approaches, since each pixel captures only the red (R), green (G) or blue (B) channel of the raw data in a spatial layout, both $\mathbf{X}$ and $\mathbf{M}$ are divided into four parts to obtain four sub-measurements $\{\mathbf{Y}^r, \mathbf{Y}^{g_1}, \mathbf{Y}^{g_2}, \mathbf{Y}^b\} \in \mathbb{R}^{\frac{H}{2} \times \frac{W}{2}}$.

$$\mathbf{Y}^r = \mathbf{X}^r \odot \mathbf{M}^r, \qquad \mathbf{Y}^{g_1} = \mathbf{X}^{g_1} \odot \mathbf{M}^{g_1}, \qquad \mathbf{Y}^{g_2} = \mathbf{X}^{g_2} \odot \mathbf{M}^{g_2}, \qquad \mathbf{Y}^b = \mathbf{X}^b \odot \mathbf{M}^b. \quad (1)$$

Some methods [14, 15] process the four sub-measurements individually, restore them to RGB color by an off-the-shelf demosaicing algorithm and finally combine them to get the final reconstructed video, which is inefficient and cannot make good use of channel correlation. Thus others [16, 17] input all sub-measurements into their proposed reconstruction network simultaneously, and finally obtain the final RGB color video by convolutional network. Similarly, we first obtain four sub-measurements based on the spatial layout of the quad-Bayer array shown in Fig. 1(a), and then feed them into the network simultaneously.

Following [11, 43], we denote the vectorized measurement $\mathbf{y} \in \mathbb{R}^{HW}$. Then given vectorized color video $\mathbf{x} \in \mathbb{R}^{HWT}$ and mask $\mathbf{\Phi} \in \mathbb{R}^{HW \times HWT}$, the degradation model can be formulated as:

$$\mathbf{y} = \mathbf{\Phi}\mathbf{x} + \mathbf{n}, \tag{2}$$

where $\mathbf{n} \in \mathbb{R}^{HW}$ represents noise on measurement. SCI reconstruction is to obtain $\mathbf{x}$ from captured $\mathbf{y}$ and the pre-set $\mathbf{\Phi}$ using a reconstruction algorithm [14, 15, 16, 44]. However, for quad-Bayer color videos, demosaicing algorithms bring artifacts, and Bayer pattern-based convolutional reconstruction causes color distortion, significantly affecting quality.

### 2.2 State Space Models (SSMs)

SSMs are common treated as linear time-invariant systems that map a 1D sequence $x(t) \in \mathbb{R}$ to $y(t) \in \mathbb{R}$ through a hidden state $h(t) \in \mathbb{R}^N$, the process can be expressed as follows:

$$\begin{aligned} h'(t) &= \mathbf{A}h(t) + \mathbf{B}x(t), \\ y(t) &= \mathbf{C}h(t), \end{aligned} \tag{3}$$

where $\mathbf{A} \in \mathbb{R}^{N \times N}, \mathbf{B} \in \mathbb{R}^{N \times 1}, \mathbf{C} \in \mathbb{R}^{1 \times N}$. It is common to use the zero-order hold (ZOH) method to discretize the continuous system, thus Eq. (3) can be discretized as following:

$$\begin{aligned} h_k &= \bar{\mathbf{A}}h_{k-1} + \bar{\mathbf{B}}x_k, & \bar{\mathbf{A}} &= e^{\triangle \mathbf{A}}, \\ y_k &= \mathbf{C}h_k, & \bar{\mathbf{B}} &= (e^{\triangle \mathbf{A}} - I)\mathbf{A}^{-1}\mathbf{B}, \end{aligned} \tag{4}$$

which uses timescale parameter $\triangle$ to convert continuous $\mathbf{A}$ and $\mathbf{B}$ to discrete $\bar{\mathbf{A}}$ and $\bar{\mathbf{B}}$.

### 2.3 SSMs for Visual Applications

The 1D S4 model [45] is extented to handle multidimensional data, while TranS4mer model [46] optimizes movie scene detection by combining S4 with self-attention. Vision Mamba and MambaIR [30, 47] introduce SSMs into the vision domain as generic backbones. U-Mamba [48] addresses long-range dependencies in biomedical images. Efficient medical image segmentation is achieved with lightM-UNet and UltraLight VM-UNet [40, 41]. ViVim [49] is proposed for effective and efficient medical video object segmentation.

These methods focus on (i) image restoration (spatial information), (ii) video understanding (global features), and (iii) medical image or video segmentation (small resolution and frame count). However, they do not apply SSMs to video SCI, a task requiring spatial-temporal consistency and detailed feature reconstruction for high-resolution, long-frame videos. Therefore, there is an urgent need to explore SSMs' performance and efficiency in long-sequence problems such as video SCI.

# 3 The Proposed Method

In this section, we introduce our proposed MambaSCI network framework, and detail our customized Residual-Mamba-Block, which is capable of capturing long-range spatial-temporal consistency along with edge detail sharpness reconstruction and channel information interaction, thus being able to outperform SOTA results with fewer parameters and FLOPS. Fig. 3 illustrates the reconstruction process, network framework and details of core modules.

## 3.1 Architecture Overview

Given the 2D measurement $\mathbf{Y} \in \mathbb{R}^{H \times W}$ and mask $\mathbf{M} \in \mathbb{R}^{H \times W \times T}$, through the general initialization module of SCI, the general initialization module of SCI provides an initial raw quad-Bayer reconstruction video $\mathbf{X}_{in} \in \mathbb{R}^{H \times W \times 1 \times T}$. This serves as the input to the MambaSCI reconstruction network, which outputs the final color reconstruction video $\mathbf{X}_{out} \in \mathbb{R}^{H \times W \times 3 \times T}$. MambaSCI comprises five main components: **(i)** shallow feature extraction block, **(ii)** encoder layer, **(iii)** bottleneck layer, **(iv)** decoder layer, **(v)** color video reconstruction block. When $\mathbf{X}_{in}$ is feed into MambaSCI, it first goes through shallow feature extraction block, which includes a depthwise separable convolution (DWConv), producing the shallow feature $\mathbf{F} \in \mathbb{R}^{H \times W \times C \times T}$. Next $\mathbf{F}$ sequentially passes through three encoder layers, each composed of a Residual-Mamba-Block and Max-Pooling operation, resulting in the feature $\mathbf{F}_{ei} \in \mathbb{R}^{(H/2^i) \times (W/2^i) \times (C \times 2^i) \times T}$, where $i \in \{1, 2, 3\}$. After these encoding layers, the deep feature $\hat{\mathbf{F}} \in \mathbb{R}^{\hat{H} \times \hat{W} \times \hat{C} \times T}$ is obtained, with $\hat{H} = \frac{H}{8}$, $\hat{W} = \frac{W}{8}$ and $\hat{C} = 8 \times C$. The bottleneck layer, composed of Residual-Mamba-Blocks, keeps the feature's shape unchanged. Then, through each decoding layer, which includes residual convolutions and upsampling operations, the feature transforms into $\mathbf{F}_{di} \in \mathbb{R}^{(\hat{H} \times 2^i) \times (\hat{W} \times 2^i) \times (\hat{C}/2^i) \times T}$. Eventually, feature $\mathbf{F}_{d3}$ is fed into the color video reconstruction block to obtain the reconstructed color video $\mathbf{X}_{out} \in \mathbb{R}^{H \times W \times 3 \times T}$. See Fig. 3(b) for an overall view.

## 3.2 Residual-Mamba-Block

Within each encoder layer, we incorporate $N$ Residual-Mamba-Blocks, specifically designed to capture and enhance temporal-spatial coherence across multiple scales, resulting in more accurate and comprehensive feature representations. As depicted in Fig. 3(c), each Residual-Mamba-Block consists of three key components: **(i)** the STMamba module, **(ii)** the EDR module, and **(iii)** the CA module. These components are detailed below, and the overall process can be mathematically described as follows:

$$\mathbf{F}_1^l = \text{STMamba}(\text{LN}(\mathbf{F}^l)) + \mathbf{F}^l \cdot s_1,$$
$$\mathbf{F}_2^l = Projection(\text{EDR}(\text{LN}(\mathbf{F}_1^l)) + \mathbf{F}_1^l \cdot s_2), \tag{5}$$
$$\mathbf{F}_{out}^l = \text{CA}(\text{LN}(\mathbf{F}_2^l)) + \mathbf{F}_2^l \cdot s_3,$$

where LN represents LayerNorm and $l \in [1, N]$, $\mathbf{F}^l$ is the $l_{th}$ block's input feature and $\mathbf{F}_{out}^l$ can be treated as $\mathbf{F}^{l+1}$ block's input, $s_1$, $s_2$, $s_3$ represent the learnable scales in the residual connection.

**(i) STMamba module.** Previous video SCI reconstruction algorithms typically compute attention separately for the temporal and spatial dimensions, then fuse them using a residual network [50]. However, this approach may lack temporal-spatial consistency. To address this issue, we employ the STMamba model [49], which integrates spatial-temporal information through structured SSMs. Specifically, As illustrated in Fig. 3(d), the input $\mathbf{F} \in \mathbb{R}^{H \times W \times C \times T}$ is processed in two parallel branches. In the first branch, $\mathbf{F}$ is expanded to $\hat{C} = S \times C$ channels via a linear layer, resulting in $\mathbf{X} \in \mathbb{R}^{H \times W \times \hat{C} \times T}$, where $S$ is expansion scale. Then, $\mathbf{X}$ is unfolded along frames $T$ to form $\mathbf{X}_s \in \mathbb{R}^{T(HW) \times \hat{C}}$. Forward and backward scanned features, $\mathbf{X}sf$ and $\mathbf{X}_{sb}$, are obtained by scanning $\mathbf{X}_s$ in both directions, efficiently capturing spatial dependencies. Simultaneously, sequence $\mathbf{X}_t \in \mathbb{R}^{(HW)T \times \hat{C}}$ is generated to explore temporal dependencies by forward scanning each pixel across the $T$ frames.

STMamba utilizes parallel SSMs to capture intra-frame and inter-frame correlations, enforcing time-space consistency constraints. This approach enables STMamba to capture both the spatial features within frames and the temporal dependencies between frames, accurately modeling dynamic changes in video data, facilitating efficient spatial-temporal feature extraction and preservation of

Table 1: Reconstruction quality and computational complexity for different versions of MambaSCI.

| Method | #Channel | Block | PSNR (dB) | SSIM | Params (M) | FLOPS (G) |
|---|---|---|---|---|---|---|
| MambaSCI-T | 8 | [2,4,4,6] | 32.13 | 0.919 | 1.61 | 165.47 |
| MambaSCI-S | 10 | [2,4,4,6] | 34.53 | 0.950 | 2.47 | 247.53 |
| MambaSCI-B | 16 | [2,4,4,6] | 35.70 | 0.959 | 6.11 | 556.89 |

consistency. As shown in Fig. 3(d), the process can be formulated as:

$$\mathbf{X} = SiLU(Linear(\mathbf{F})),$$
$$\mathbf{Z} = SiLU(Linear(\mathbf{F})),$$
$$\mathbf{X}_s, \mathbf{X}_t = \text{Unfolding}(\mathbf{X}),$$
$$\mathbf{X}_{sf} = LN(\text{Forward-SSM}(Conv1d(\mathbf{X}_s))),$$
$$\mathbf{X}_{sb} = LN(\text{Backward-SSM}(Conv1d(\mathbf{X}_s))),$$
$$\mathbf{X}_t = LN(\text{Forward-SSM}(Conv1d(\mathbf{X}_t))),$$
$$\mathbf{F}_{ssm} = Linear(\mathbf{X}_{sf} \odot \mathbf{Z} + \mathbf{X}_{sb} \odot \mathbf{Z} + \mathbf{X}_t \odot \mathbf{Z}),$$

(6)

where $\odot$ represents Hadamard product, LN represents LayerNorm and $SiLU$ is an activation function.

**(ii) EDR and CA module.**

To achieve the fine reconstruction of edge details and to compensate for the missing inter-channel interaction capability in the Mamba model, we introduce the EDR and CA modules. The EDR module consists of linear layers and depthwise separable convolution (DW-Conv). By combining linear transformation with DWConv features, the EDR module gains the ability to perform multi-scale feature fusion, allowing the network to extract global information from local features and process both global and local information simultaneously. This enhances the model's capacity to understand complex edge structures. Additionally, through adaptive weight initialization,

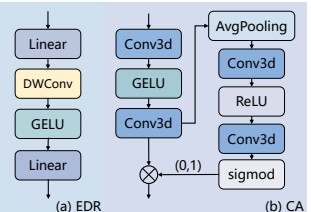

Figure 4: Detailed design of EDR and CA module.

the model can capture finer details more effectively in the early stages of training, enabling efficient edge detail reconstruction in a lightweight manner while enhancing image edge information.

On the other hand, the CA module compresses the spatial dimension of the feature map to $1 \times 1 \times 1$ through an average pooling operation, which in turn maps the compressed features to the interval $(0, 1)$ through a convolutional layer with an activation function to form channel attention weights. These weights are subsequently multiplied channel-by-channel with the original feature map to implement the channel attention mechanism. In addition, the CA module utilizes the convolution operation to further facilitate the fusion and exchange of information between channels, thereby enhancing the interaction effect between channels.

## 3.3 Bottleneck and Decoder Layer.

Like transformer, Mamba encounters severe optimization and convergence challenges as network depth increases [40]. In the bottleneck layer, multiple Residual-Mamba-Blocks are concatenated, keeping the same number of feature channels and resolution. This maintains feature richness and clarity, enhances the model's spatial-temporal dependency capture, and improves MambaSCI's performance without significantly increasing computational burden.

Decoding layer decodes features and recover image resolution. It receives two inputs: $\mathbf{F}_e \in \mathbb{R}^{\hat{H} \times \hat{W} \times \hat{C} \times T}$ from the skip connection, which retains original spatial information, and $\mathbf{F}_d \in \mathbb{R}^{\hat{H} \times \hat{W} \times \hat{C} \times T}$ from the previous decoding layer, containing higher-level spatial-temporal information. The decoding layer fuses features using element-wise addition to enhance expressiveness, applies DWConv with residual concatenation, an activation function and an upsampling operation. This process produces output features with richer semantics and higher spatial resolution.

## 3.4 Color Video Reconstruction Block.

The color video reconstruction block reconstructs the desired video $\mathbf{X}_{out} \in \mathbb{R}^{H \times W \times 3 \times T}$. Instead of the computationally intensive remosaicing and demosaicing of raw quad-Bayer images, we use a

Table 2: Comparisons between MambaSCI and SOTA methods on 6 simulation videos. PSNR (upper entry in each cell), and SSIM (lower entry in each cell) are reported. The best and second-best results are highlighted in bold and underlined, respectively.

| Method | Params (M) | FLOPS (G) | Beauty | Bosphours | Ruuner | ShakeNDry | Traffic | Jockey | Avg |
|---|---|---|---|---|---|---|---|---|---|
| GAP-TV [13] | - | - | 33.38
0.965 | 29.53
0.904 | 29.61
0.872 | 29.70
0.884 | 19.64
0.625 | 29.32
0.885 | 28.53
0.856 |
| PnP-FFDnet-gray [14] | - | - | 32.47
0.958 | 27.45
0.883 | 28.66
0.864 | 26.93
0.832 | 20.56
0.686 | 31.07
0.9.6 | 27.86
0.855 |
| PnP-FastDVD-gray [15] | - | - | 34.29
0.967 | 33.07
0.947 | 34.18
0.928 | 30.11
0.883 | 23.74
0.811 | 32.70
0.921 | 31.35
0.909 |
| EfficientSCI-S [16] | 2.21 | 1434.18 | 19.47
0.402 | 26.88
0.642 | 34.26
0.906 | 24.13
0.639 | 25.98
0.761 | 30.41
0.788 | 26.86
0.690 |
| EfficientSCI-B [16] | 8.83 | 5701.50 | 36.40
_0.980_ | 24.52
0.497 | 36.34
0.919 | _34.73_
_0.955_ | 26.63
0.774 | 35.52
0.945 | 32.35
0.845 |
| STFormer-S [17] | **1.23** | 769.23 | 23.15
0.679 | 23.75
0.435 | 34.36
0.885 | 24.78
0.659 | 26.17
0.771 | 30.13
0.785 | 27.06
0.703 |
| STFormer-B [17] | 19.49 | 12155.47 | _36.69_
**0.981** | 23.84
0.446 | 37.13
0.927 | **34.83**
**0.955** | 26.62
0.791 | _35.80_
_0.952_ | 32.48
0.842 |
| **MambaSCI-T** | 1.61 | **165.47** | 33.45
0.965 | 35.07
0.963 | 35.03
0.926 | 31.81
0.912 | 25.31
0.843 | 32.09
0.909 | 32.13
0.919 |
| **MambaSCI-S** | 2.47 | 247.53 | 36.12
0.978 | _37.33_
_0.976_ | _38.35_
_0.968_ | 33.72
0.943 | _26.70_
_0.886_ | 34.98
0.951 | _34.53_
_0.950_ |
| **MambaSCI-B** | 6.11 | 556.89 | **36.95**
0.979 | **38.62**
**0.982** | **40.02**
**0.977** | 34.55
0.950 | **27.52**
**0.904** | **36.54**
**0.960** | **35.70**
**0.959** |

three-layer convolution (kernel sizes $3\times3\times3$, $3\times3\times3$, and $1\times1\times1$) to process the decoding layer's output and obtain the final RGB color video.

## 3.5 Network Variants and Computational Complexity

To balance size and performance, we propose three versions of the MambaSCI model: MambaSCI-T (*tiny*), MambaSCI-S (*small*), and MambaSCI-B (*base*). Tab. 1 shows the network hyperparameters, model parameters, and computational complexity (FLOPS). By varying the number of channels from the initial DWConv, our method achieves significantly lower complexity than EfficientSCI [16] and STFormer [17].

Table 3: Computational complexity of several SOTAs.

| Method | Computational Complexity |
|---|---|
| STFormer | $6HWTC^2 + 2G_hG_wHWTC + HWT^2C$ |
| EfficientSCI | $\frac{1}{2}HWTK^2C^2 + \frac{1}{2}HWTC^2 + \frac{1}{2}HWT^2C$ |
| **MambaSCI** | $8HWTCN + 2HWTCN^2$ |

We also calculate the computational complexity of the attention module n MambaSCI compared to other SOTA methods, as shown in Tab. 3, where $C$ is the number of input features, $K$ represents the kernel size, $G_h$ and $G_w$ are the spatial size of local window in Swin-transformer [51], $N$ is a fixed parameter in Mamba set to 16. In MambaSCI, input features are unfolded into a sequence $\mathbf{S} \in \mathbb{R}^{HW \times C \times T}$. As seen in Tab. 3, while STFormer and EfficientSCI scale linearly with the spatial size ($HW$), their complexity grows quadratically with video frames $T$ and $C$, which is typically 64 or larger, resulting in high computational costs. MambaSCI scales linearly with the entire video sequence ($HWT$) and $C$, which is capped at 64 in MambaSCI-B, enabling efficient reconstruction of longer video sequences. Inference time comparisons are shown under various methods in Tab. 5.

# 4 Experiment

In this section, we evaluate MambaSCI against SOTA video reconstruction methods on multiple simulation datasets using PSNR, SSIM metrics, and visual comparisons.

## 4.1 Experimental Setup

Following STFormer and EfficientSCI, we use DAVIS2017 [52] with resolution $480\times894$ (480p) as the model training dataset. To verify model performance, we test our MambaSCI on several

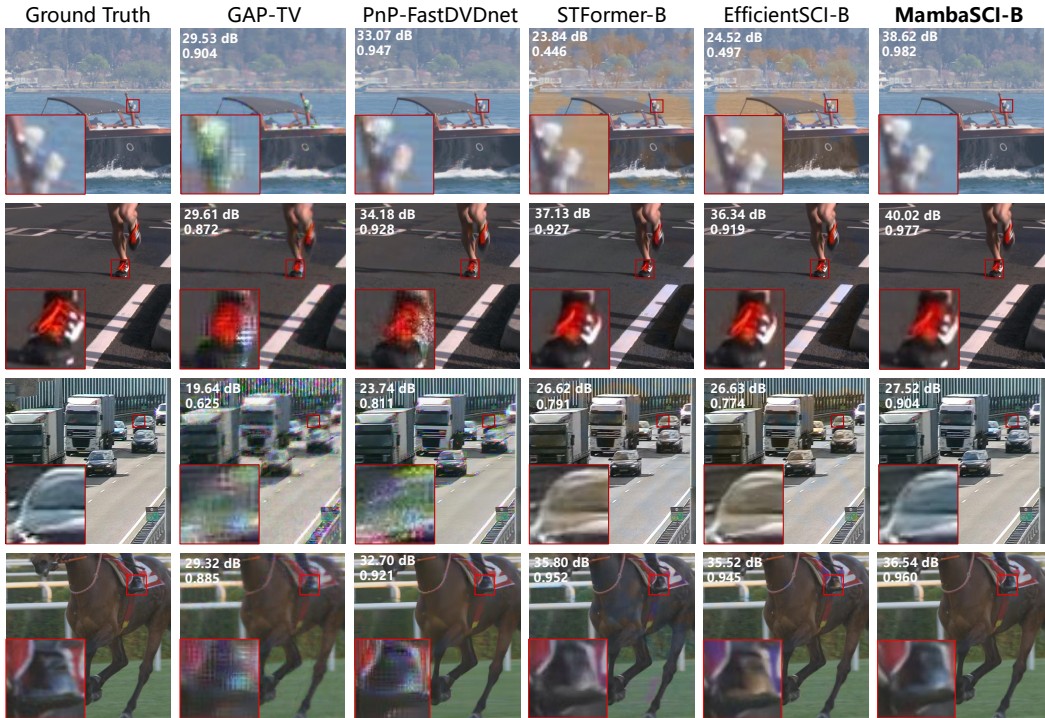

Figure 5: Visual reconstruction results of different algorithms on middle-scale simulation color video dataset (`Bosphrous #10`, `Runner #11`, `Traffic #32` and `Jockey #24` in order from top to bottom). PSNR/SSIM is shown in the upper left corner of each picture.

simulated datasets, six benchmark mid-scale color datasets [15] (`Beauty`, `Bosphorus`, `Jockey`, `Runner`, `ShakeNDry` and `Traffic` of size 512×512×3×32), and four benchmark large-scale color datasets [15] (`Messi`, `Hummingbird`, `Swinger` and `Football`). Since there is currently no real color video SCI dataset based on quad-Bayer pattern, our method is not tested on real datasets.

## 4.2 Implementation Details

We use PyTorch framework training on 4 NVIDIA RTX4090 GPUs and use random flipping, scaling, and cropping on `DAVIS2017` for data augmentation. We use randomly generated masks as training input to enhance model robustness and optimize the model using the Adam [53] optimizer. Since MambaSCI is flexible in input size, we first train for 100 epochs at a learning rate of 0.0005 on data with a spatial size of 128×128. Then, we train for 50 epochs at learning rate of 0.0001, followed by fine-tuning on 256×256 data at learning rate of 0.00001 for an additional 50 epochs.

## 4.3 Results on Middle-scale Simulation Color Video

To test the performance of our method for color video reconstruction, we perform experiments on a 32-frame simulation color RGB video dataset with size of $512 \times 512 \times 3 \times 32$. We compress the color video with compression rate of $B = 8$ and capture quad-Bayer pattern measurements using a camera with quad-Bayer CFA pattern.

Since all current color video SCI reconstruction algorithms are designed based on Bayer pattern, which cannot be directly applied to quad-Bayer pattern. Meanwhile, re-training an E2E model requires much training time and memory. Thus we only re-train two of the latest SOTA E2E models (STFormer [17], EfficientSCI [16]) with quad-Bayer pattern. We compare with iterative optimization algorithm (GAP-TV [13]), PnP algorithms (PnP-FFDnet [14] and PnP-FastDVD [15]) and E2E algorithms (STFormer [17] and EfficientSCI [16]). The number of parameters, FLOPS, and reconstruction results are shown in Tab. 2.

Notably, there is no readily available quad-Bayer demosaicing package. Therefore, for model-based and PnP algorithms, we first upsample the reconstructed raw quad-Bayer video $\mathbf{X} \in \mathbb{R}^{H \times W \times 1 \times T}$

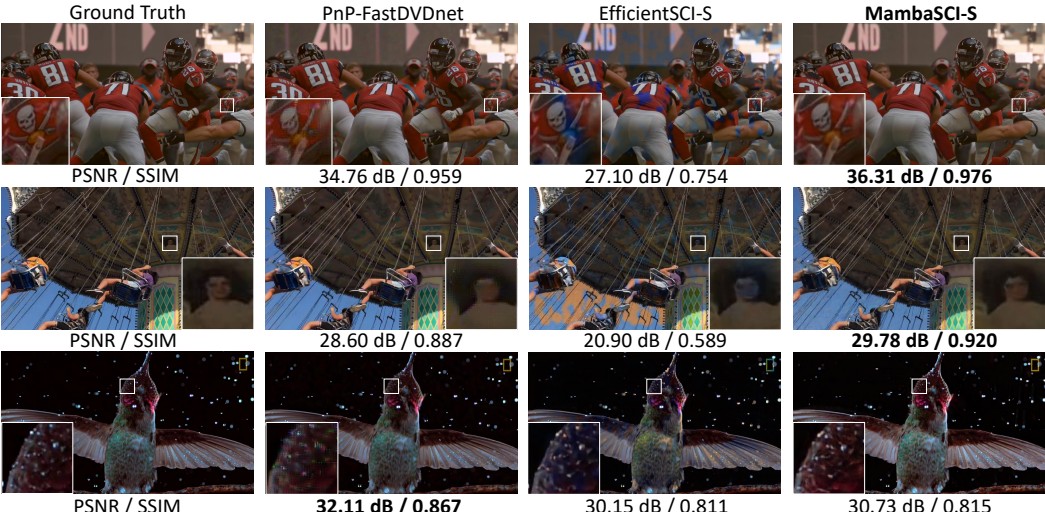

| Ground Truth | PnP-FastDVDnet | EfficientSCI-S | **MambaSCI-S** |
|---|---|---|---|
| PSNR / SSIM | 34.76 dB / 0.959 | 27.10 dB / 0.754 | **36.31 dB / 0.976** |
| PSNR / SSIM | 28.60 dB / 0.887 | 20.90 dB / 0.589 | **29.78 dB / 0.920** |
| PSNR / SSIM | **32.11 dB / 0.867** | 30.15 dB / 0.811 | 30.73 dB / 0.815 |

Figure 6: Visual reconstruction results of different algorithms on large-scale simulated color video dataset (`Footbal #11`, `Swinger #1`, and `Hummingbird #40` from top to bottom). PSNR/SSIM is under each image.

according to the quad-Bayer CFA pattern to obtain a three-channel video $\hat{\mathbf{X}} \in \mathbb{R}^{H \times W \times 3 \times T}$. We then demosaic it using the BJDD [54] algorithm to get the final RGB color video. See supplementary materials for details. Visual comparisons of the reconstruction results are shown in Fig. 5.

We summarize our observations: **(i)** Our MambaSCI model significantly outperforms SOTA methods with lower computational and memory resources. For instance, MambaSCI-B surpasses STFormer-B by 3.22 dB, using only **31%** (6.11 / 19.49) of the Params and **4.5%** (556.89 / 12155.47) of the FLOPS. It also outperforms EfficientSCI-B by 3.35 dB with just **69%** Params and **9.8%** FLOPS. Additionally, both MambaSCI-S and MambaSCI-T achieve better results than STFormer-S and EfficientSCI-S with fewer Params and FLOPS. Fig. 1(b) shows MambaSCI's superior PSNR-FLOPS performance with fewer resources. **(ii)** In visual comparisons, GAP-TV and PnP-based methods exhibit artifacts, while STFormer and EfficientSCI suffer from color distortions likely because their reconstruction modules being designed for Bayer pattern and not compatible with Quad-Bayer pattern. None of these methods achieve high-quality reconstruction. MambaSCI, however, eliminates artifacts and achieves high-quality reconstruction with accurate color fidelity.

In addition, our proposed model demonstrates SOTA reconstruction performance even at higher frame rates and compression ratios. We tested it on *beauty* data across various compression ratios (B = 8, 16, 32), with results detailed in the Tab. 4. Notably, our method requires only 9.8% of the FLOPS needed by EfficientSCI, while also significantly outperforming it in the SSIM.

Table 4: Performance analysis at $B$=16 and 32 cases.

| B | Methods | Params (M) | FLOPS (G) | PSNR (dB) | SSIM |
|---|---|---|---|---|---|
| 16 | PnP-FFDnet | - | - | 24.85 | 0.767 |
| | STFormer | 19.49 | 24311.76 | 25.21 | 0.685 |
| | EfficientSCI | 8.83 | 11406.23 | 25.35 | 0.656 |
| | MambaSCI | **6.11** | **1113.78** | 25.39 | **0.817** |
| 32 | PnP-FFDnet | - | - | 1.82 | 0.496 |
| | STFormer | 19.49 | OOM | - | - |
| | EfficientSCI | 8.83 | 22825.34 | **23.24** | 0.653 |
| | MambaSCI | **6.11** | **2227.57** | 22.44 | **0.785** |

## 4.4 Results on Large-scale Simulation Color Video

Similar to previous studies, we conduct experiments on a large-scale color video dataset. Due to the significant time and memory required to retrain existing E2E models for Bayer patterns, we only compare with SOTA model-based methods (GAP-TV, PnP-FFDnet, PnP-FastDVDnet) and retrain STFormer-S and EfficientSCI-S for the quad-Bayer pattern. Tab. 5 shows the comparisons on PSNR and SSIM, and Fig. 6 provides visual comparisons.

We summarize the observations: **(i)** MambaSCI-S outperforms other methods in PSNR and SSIM on `football` and `swinger`, achieving over 1.5 dB higher PSNR on `football`. **(ii)** In visual comparisons, GAP-TV and PnP algorithms exhibit artifacts, while STFormer and EfficientSCI suffer from color distortions. MambaSCI excels in reconstructing detailed information, resulting in superior reconstruction quality. **(iii)** The poorer performance on `Messi` and `Hummingbird` may be due to faster, more detailed motions that strained by limited parameters and FLOPS. However, the visual results remain superior to other SOTA methods.

Table 5: Comparisons between MambaSCI and SOTA methods on 4 large-scale simulation videos. PSNR (upper), and SSIM (lower) are reported. The total time (minutes) taken to reconstruct 4 videos is under each method. The best and second-best results are highlighted in bold and underlined.

| Dataset | Pixel resolution | GAP-TV (17.03) | PnP-FFDNET (17.97) | PnP-FastDVDnet (50.03) | STFormer-S (**2.12**) | EfficientSCI-S (5.47) | **MambaSCI-S** (_4.98_) |
|---------|-----------------|----------------|--------------------|------------------------|----------------------|-----------------------|-------------------------|
| Messi | $1080 \times 1920 \times 3 \times 48$ | 25.00 0.868 | _28.62_ _0.939_ | **29.17** **0.939** | 17.77 0.639 | 18.45 0.685 | 26.36 0.874 |
| Hummingbird | $1080 \times 1920 \times 3 \times 40$ | 29.33 0.886 | 29.72 **0.924** | **32.11** _0.867_ | _31.96_ 0.886 | 30.15 0.811 | 30.73 0.815 |
| Swinger | $1080 \times 1920 \times 3 \times 20$ | 24.92 0.833 | 26.72 0.883 | _28.60_ _0.887_ | 20.10 0.556 | 20.90 0.589 | **29.78** **0.920** |
| Football | $1080 \times 1920 \times 3 \times 48$ | 31.19 0.939 | 33.82 0.963 | _34.76_ _0.959_ | 30.61 0.815 | 27.10 0.754 | **36.31** **0.976** |

## 4.5 Ablation Study

We conduct ablation experiments to evaluate the effectiveness of each module in MambaSCI. Tab. 6 presents the results, comparing reconstruction quality, Params, and FLOPS across different models. All experiments are performed on six color benchmark datasets.

**STMamaba Block:** We verify the impact of STMamba blocks on reconstruction quality. As indicated in Tab. 6, replacing vanilla Mamba with STMamba boosts PSNR by 6dB, while Params and FLOPS remain unchanged. STMamba's linear scanning enables satisfying spatial-temporal consistency without a notable increase in computational complexity.

**EDR Block:** Tab. 6 demonstrates that the EDR module can improve PSNR by approximately 4.3dB, dramatically improving the quality of the reconstruction. However, it will result in the increase of Params and FLOPS.

Table 6: Ablation study on each major module.

| Baseline | STMamba | EDR | CA | PSNR | SSIM | Params(M) | FLOPS(G) |
|----------|---------|-----|----|------|------|-----------|----------|
| ✓ | | | | 24.18 | 0.811 | 0.28 | 35.36 |
| ✓ | ✓ | | | 30.31 | 0.897 | 0.28 | 35.36 |
| ✓ | ✓ | ✓ | | 34.66 | 0.953 | 2.53 | 235.47 |
| ✓ | ✓ | ✓ | ✓ | **35.70** | **0.959** | 6.11 | 556.89 |

**CA Block:** CA block compensates for the lack of channel information interaction in Mamba model, achieves the modelling of channel importance and improves the reconstruction quality through channel attention mechanism. Tab. 6 shows CA block can significantly improve reconstruction quality. However, the CA module's multiple $Conv3d$ operations result in a notable increase in parameters and FLOPS, which is an aspect to optimize in future work.

**Number of Channels:** Tab. 1 illustrates that the only distinction among different MambaSCI versions is the varying number of channels. Through experimentation, we observed that Params and FLOPS are significantly influenced by

Table 7: Ablation study on Residual-Mamba-Block

| Models | PSNR (dB) | SSIM | Params (M) | FLOPS (G) |
|--------|-----------|------|------------|-----------|
| w/o learnable scale | 35.33 | 0.955 | 6.11 | 556.89 |
| w/o residual connections | 34.71 | 0.953 | 6.11 | 556.89 |
| **Residual-STMamba-Block** | **35.70** | **0.959** | 6.11 | 556.89 |

the number of channels, rather than the number of Residual-Mamba-Blocks. Moreover, an excess of Residual-Mamba-Blocks prolongs both training and inference times, highlighting the need for a trade-off in the current setup.

**Residual Connection and Learnable Scales:** The Residual-STMamba-Block is the core customised module of our MambaSCI. Tab. 7 is experimentally demonstrated that the residual connections and the learnable scales are effective in improving the reconstruction quality enhancement.

## 5 Conclusion

In this paper, we introduced quad-Bayer pattern into video SCI for the first time, enabling SCI to align with the fact that most current videos are captured by mobile phones with quad-Bayer cameras, thus avoiding artifacts and color distortions caused by existing algorithms. Specifically, we integrate Mamba model with an asymmetric UNet in video SCI, leveraging Mamba's linear complexity and the speed improvements from the non-symmetric architecture for efficient SCI reconstruction. Moreover, we customized Residual-Mamba-Blocks to connect STMamba, EDR, and CA modules through residual connections, ensuring efficient spatial-temporal consistency and detailed reconstruction. Experimental results on simulated color video datasets highlighted that MambaSCI outperformed SOTA methods with fewer parameters, lower computational complexity and better visual effects.

## Acknowledgments

This work was supported in part by the Shenzhen Science and Technology Innovation Committee under Grant JSGG20220831104402004 and by Guangdong Major Project of Basic and Applied Basic Research under Grant 2023B0303000010.

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

# A  Appendix

In the supplementary material, we provide more details that are not in out main paper:

**(a)** Mathematical model of color video CACTI in Sec. A.1.

**(b)** The pseudo-code of Residual-Mamba-Block in Sec. A.2.

**(c)** We apply demosaicing techniques to raw quad-Bayer images reconstructed by GAP-TV, PnP-FFDNet, and PnP-FastDVDnet, converting them into RGB images for visual comparison. Meanwhile, more visual comparison against the current state-of-art (SOTA) method both on middle-scale and large-scale simulation color videos in Sec. A.3.

**(d)** Limitation of our work in Sec. A.4.

**(e)** Broader impacts in Sec. A.5

## A.1   Mathematical model of color video CACTI

Coded aperture compressive temporal imaging (CACTI) is one of the famous video SCI systems. Specifically, for color video SCI systems utilizing quad-Bayer arrays, the raw data is spatially structured that each pixel captures only red (R), green (G), or blue (B), creating a format similar to 'RGGB' where each color occupies adjacent $2\times2$ pixels in succession. Thus, the initial color video $\bar{\mathbf{X}} \in \mathbb{R}^{H \times W \times 3 \times T}$ is multiplied pixel by pixel with the filter and superimposed in the channel dimension, ultimately producing the raw data $\mathbf{X} \in \mathbb{R}^{H \times W \times T}$. Given the mask $\mathbf{M} \in \mathbb{R}^{H \times W \times T}$, the modulation is:

$$\mathbf{X}'(:,:,t) = \mathbf{X}(:,:,t) \odot \mathbf{M}(:,:,t), \tag{7}$$

where $\mathbf{X}'$ denotes the modulated video data and $\odot$ represents element-wise multiplication. 2D compressed measurement $\mathbf{Y}$ can be captured across time dimention, which can be expressed as:

$$\mathbf{Y} = \sum_{t=1}^{T} \mathbf{X}'_t + \mathbf{N}, \tag{8}$$

where $\mathbf{N} \in \mathbb{R}^{H \times W}$ means the measurement noise.

**Vectorization.** Given:

$$\mathbf{y} = \text{vec}(\mathbf{Y}) \in \mathbb{R}^{HW}, \tag{9}$$

$$\mathbf{n} = \text{vec}(\mathbf{N}) \in \mathbb{R}^{HW}, \tag{10}$$

$$\mathbf{x} = [\mathbf{x}_1^T, ..., \mathbf{x}_T^T]^T \in \mathbb{R}^{HWT}, \tag{11}$$

$$\mathbf{\Phi} = [\mathbf{D}_1, ..., \mathbf{D}_T] \in \mathbb{R}^{HW \times HWT}, \tag{12}$$

where $\text{vec}(\cdot)$ means vectorization operation, $\mathbf{x}_t = \text{vec}(\mathbf{X}(:,:,t))$ represents the vectorization of $t$-th frame of $\mathbf{X}$ and $\mathbf{D}_t = \text{Diag}(\text{vec}(\mathbf{M}(:,:,t)))$ is a diagonal matrix and its diagonal elements is filled by $\text{vec}(\mathbf{M}(:,:,t))$. The vectorization forward process can be expressed as:

$$\mathbf{y} = \mathbf{\Phi}\mathbf{x} + \mathbf{n}. \tag{13}$$

And the reconstruction process is to investigate algorithms that can reconstruct $\mathbf{x}$ given $\mathbf{y}$ and $\mathbf{\Phi}$.

## A.2   Pseudo-code of Residual-Mamba-Block

As mentioned in the paper. Residual-Mamba-Blocks can complete high quality reconstruction by capturing spatial-temporal coherence while completing edge detail reconstruction and compensating for channel information interactions not available in the Mamba model. In Algorithm 1, we show the PyTorch style pseudo-code on how to construct a Residual-Mamba-Block.

## A.3   Additional Visual Results

In this section, we will demonstrate how we employ certain techniques to apply existing demosaicing algorithms to the raw quad-Bayer images reconstructed by GAP-TV, PnP-FFDNet, and

**Algorithm 1** Pseudo-code of Residual-Mamba-Block

```python
class RSTMamba():

    def __init__(self, input_dim, output_dim, nframes, d_state = 16,
    d_conv = 4, expand = 2, mlp_ratio = 4, act_layer = nn.GELU):
        self.norm1 = LayerNorm(input_dim)
        ## STMamba
        self.mamba = Mamba(
                d_model = input_dim,  ## Model dimension d_model
                d_state = d_state,  ## SSM state expansion factor
                d_conv = d_conv,  ## Local convolution width
                expand = expand,  ## Block expansion factor
                nframes = nframes  ## Number of frames of video
                )
        self.proj = Linear(input_dim, output_dim)
        self.scale1 = nn.Parameter(torch.ones(1))
        self.scale2 = nn.Parameter(torch.ones(1))
        self.scale3 = nn.Parameter(torch.ones(1))
        self.norm2 = LayerNorm(input_dim)
        self.norm3 = LayerNorm(output_dim)
        ## Channel Attention
        self.ca = CA(output_dim)
        mlp_hidden_dim = imnt(input_dim * mlp_ratio)
        ## Edge Detail Reconstruction
        self.edr = Mlp(input_dim, mlp_hidden_dim, drop)

    def forward(self, x):
        ## x in shape [B, C, T, H, W]
        B, C, nf, H, W = x.shape
        n_tokens = x.shape[2:].numel()
        img_dims = x.shape[2:]
        x_flat = x.reshape(B, C, n_tokens).transpose(-1, -2)
        ## x_flat in shape [B, n_tokens, C]

        ## (1) STMamba and residual connection
        x_mamba = x_flat * self.scale1 + sef.drop(self.mamba(self.norm1(x_flat)))
        ## (2) EDR and residual connection
        x_mamba = x_mamba * self.scale2 +
                self.drop(self.edr(self.norm2(x_mamba), nf, H, W))
        x_mambna = self.proj(x_mamba)
        out = out.permute(0,2,3,4,1)
        ## (3) CA and residual connection
        out = out * self.skip_scale3 +
        self.ca(self.norm3(out).permute(0,4,1,2,3)).permute(0,2,3,4,1)
        out = out.permute(0,4,1,2,3)

        return out
```

PnP-FastDVDnet. This process converts the raw format images into RGB color images, facilitating visual comparison. ***We provide moving images in GIF format for the various methods in Tab. 2. Please refer to the folder 'gif' for further observation.***

**Demosaicing Process.** As mentioned in the paper, previous model-based and PnP reconstruction algorithms need to be paired with corresponding demosaicing algorithm to get RGB color videos. However there are no available packages to use for quad-Bayer demosaicing, so the problem of how to present the reconstructed color images is also addressed. As shown in Fig. 7(a), it's a video frame in raw format obtained after reconstruction, we use Algorithm. 2 to transform it into the form of Fig. 7(b) and then use the off-the-shelf demosaicing algorithm to get the final color video frame shown in Fig. 7(c).

**Algorithm 2** Code of Transformation

```python
def mask_CFA_quad(shape: int) -> Tuple[NDArray, ...]:
    channels = {channel: np.zeros(shape, dtype="bool") for channel in range(3)}

    ## Quad R (red)
    channels[0][::4,::4] = 1
    channels[0][1::4,1::4] = 1
    channels[0][::4,1::4]=1
    channels[0][1::4,::4] =1

    ## Quad G1 (grenn)
    channels[1][::4,2::4] =1
    channels[1][::4,3::4] =1
    channels[1][1::4,2::4] =1
    channels[1][1::4,3::4] =1

    ## Quad G2 (grenn)
    channels[1][2::4,::4] =1
    channels[1][2::4,1::4] =1
    channels[1][3::4,::4] =1
    channels[1][3::4,1::4] =1

    ## Quad B (blue)
    channels[2][2::4,2::4] = 1
    channels[2][3::4,2::4] = 1
    channels[2][2::4,3::4] = 1
    channels[2][3::4,3::4] = 1

    return tuple(channels.values())

def CFA_quad (CFA):
    CFA = as_float_array(CFA)
    R_m, G_m, B_m = masks_CFA_Bayer(CFA.shape)
    ## obtain red channel
    R = CFA * R_m
    ## obtain green channel
    G = CFA * G_m
    ## obtain blue channel
    B = CFA * B_m
    RGB = tstack([B, G, R])
    return RGB
```

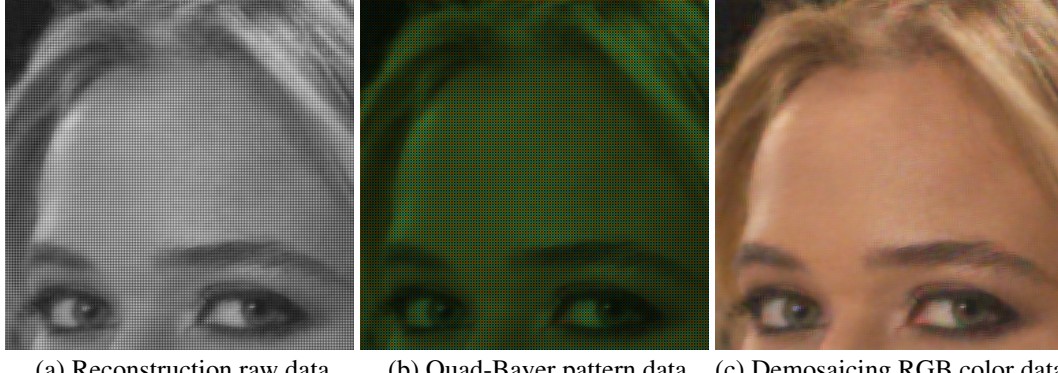

(a) Reconstruction raw data  (b) Quad-Bayer pattern data  (c) Demosaicing RGB color data

Figure 7: Process of reconstruction from Raw to color RGB image.

**Visual comparison.** We conduct additional comparative experiments with current with current SOTA methods and provide more visual comparison. As shown in Fig. 8, Fig. 9 and Fig. 10, our proposed

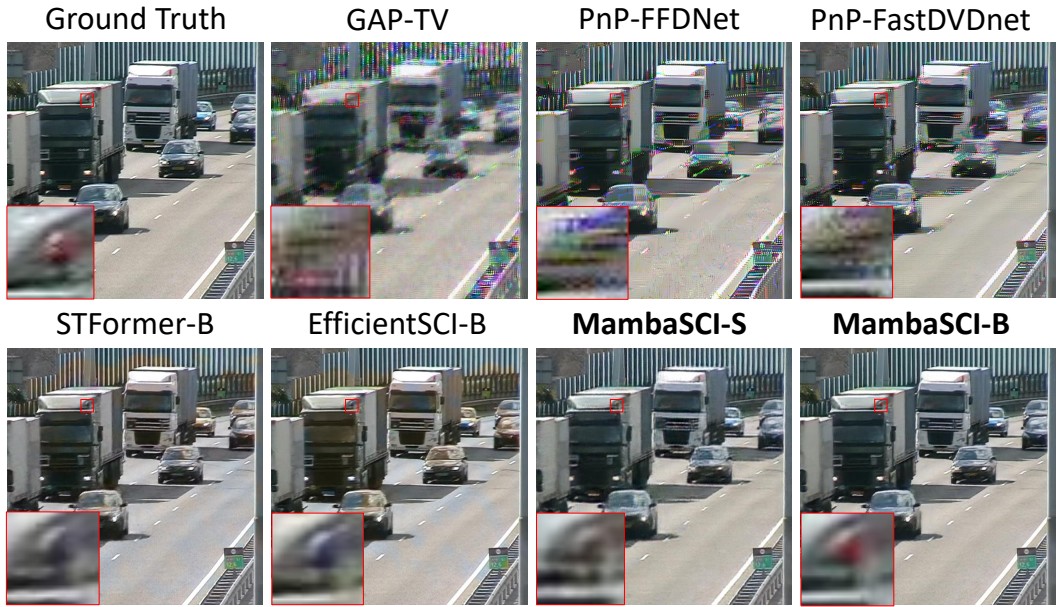

Figure 8: Visual reconstruction results of different algorithms on middle-scale simulation color video `Traffic` #16.

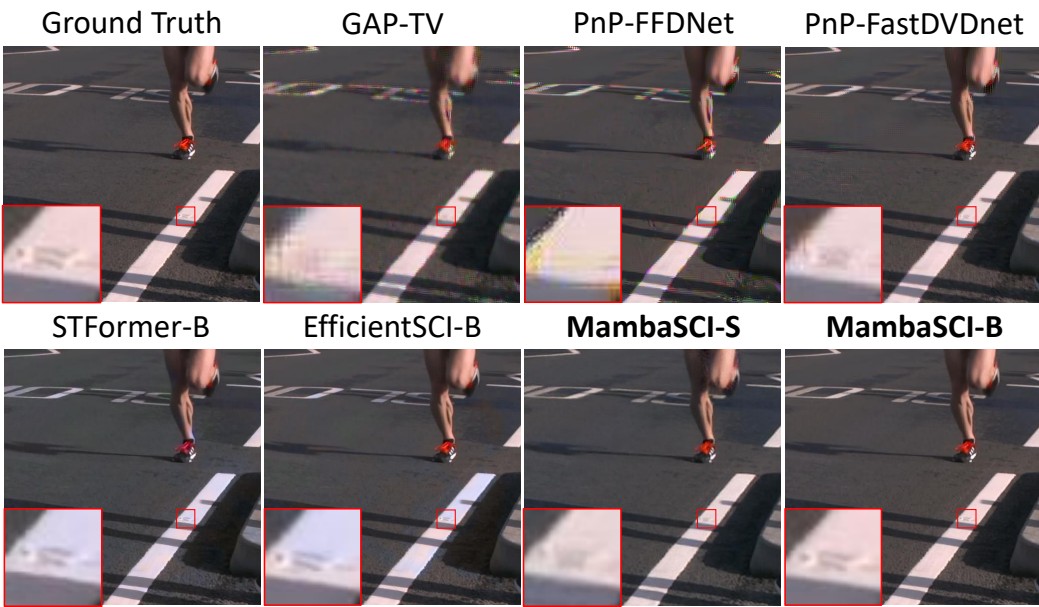

Figure 9: Visual reconstruction results of different algorithms on middle-scale simulation color video `Runner` #7.

MambaSCI method achieves superior color fidelity and detailed reconstruction on middle-scale benchmark simulation color video dataset, offering a significant visual advantage over previous methods. Meanwhile, as shown in Fig. 11, even though our proposed MambaSCI method may not surpass PnP-FastDVDFnet in terms of PSNR and SSIM metrics, it achieve superior reconstruction visual quality. In comparison to the artifacts introduced by GAP-TV and PnP-FastDVDFnet as well as the color distortions in STFormer-S and EfficientSCI-S, MambaSCI demonstrate significantly better visual results. As shown in Fig. 12 and Fig. 13, our proposed method achieve pleasant visual results.

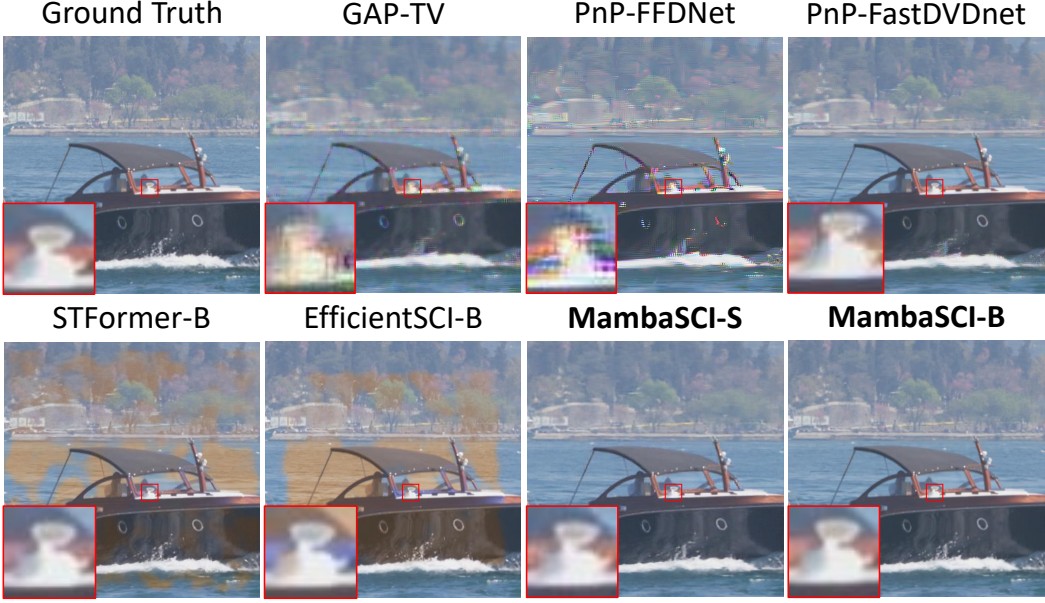

Figure 10: Visual reconstruction results of different algorithms on middle-scale simulation color video `Bosphrous #16`.

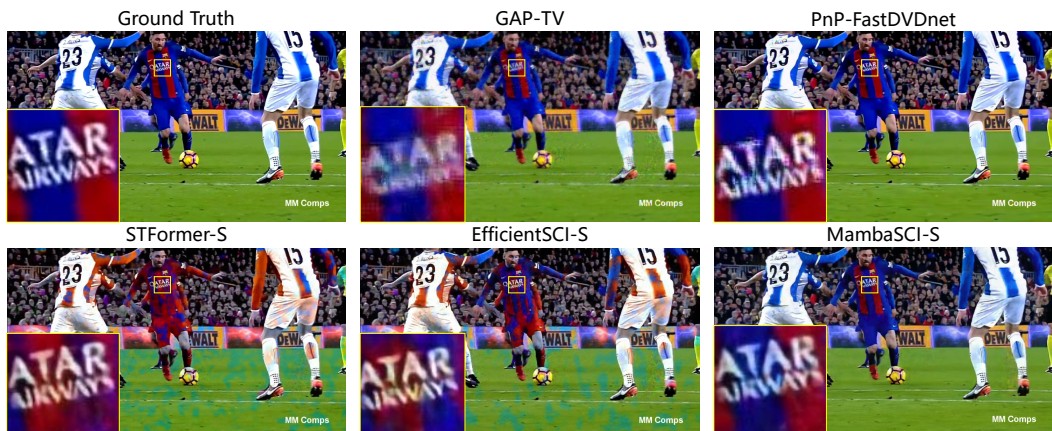

Figure 11: Visual reconstruction results of different algorithms on large-scale simulation color video `Messi #8`.

## A.4 Limitation.

The limitations of our approach are in two respects:

**(i)** One limitation is the trade-off between computational complexity and performance. As shown in Table 6, to achieve better performance, we incorporate EDR and CA modules. However, the use of Conv3d in the CA module significantly increases the number of parameters and computational complexity.

**(ii)** The second limitation is that measurements based on quad-Bayer in real datasets are currently unavailable, making it impossible to evaluate the performance of our proposed model in real-world scenarios.

Given these limitations, we aim to investigate ways to simplify the internal modules while maintaining performance, thereby reducing computational complexity and accelerating inference speed. Additionally, we will focus on collecting quad-Bayer-based SCI measurements in real-world scenarios to verify the reliability of our method under real scene conditions.

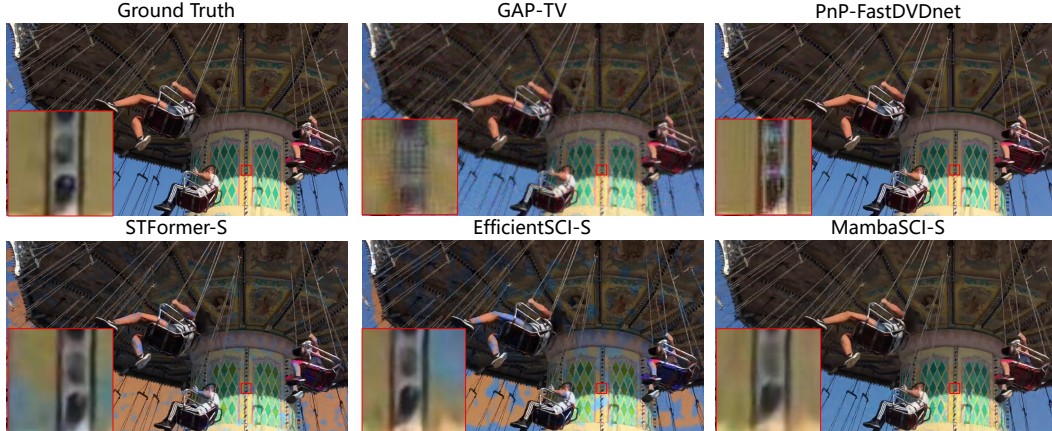

Figure 12: Visual reconstruction results of different algorithms on large-scale simulation color video `Swinger` #30.

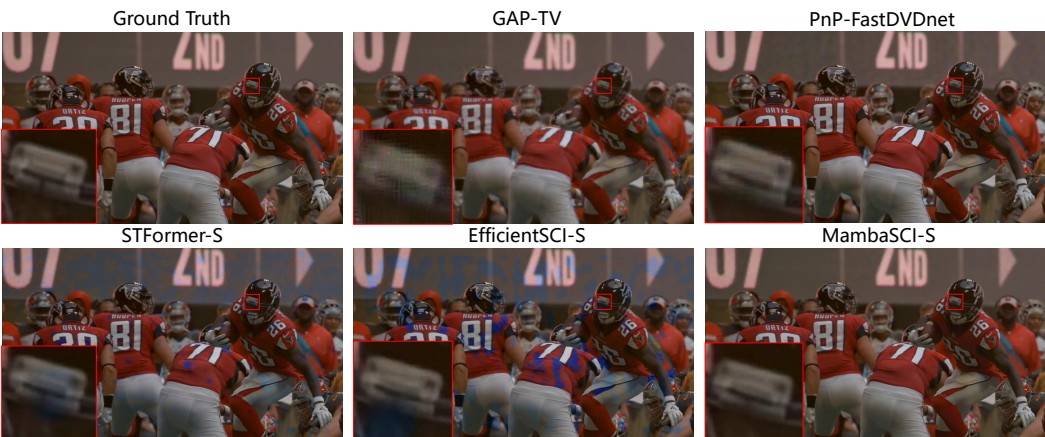

Figure 13: Visual reconstruction results of different algorithms on large-scale simulation color video `Football` #38.

## A.5  Broader Impacts

Video reconstruction is a fundamental task in snapshot compressive imaging (SCI), an area with a research history spanning several decades. As artificial intelligence continues to evolve, the handling of high-quality, high-dimensional data has emerged as a significant challenge for large-scale deep learning models. Video SCI systems, which utilize low-speed cameras to capture high-speed video, present several advantages including low memory requirements, low transmission bandwidth, low cost, and low power consumption [17, 55]. Our proposed MambaSCI algorithm enables more efficient high-quality reconstruction of videos captured with quad-Bayer pattern, significantly broadening the application scenarios of video SCI.

To date, video reconstruction techniques have not demonstrated any negative social impact. Similarly, our proposed MambaSCI algorithm does not present any foreseeable negative societal consequences.

