# OpenReview forum: "MambaSCI: Efficient Mamba-UNet for Quad-Bayer Patterned Video Snapshot Compressive Imaging"
_NeurIPS.cc/2024/Conference — NeurIPS 2024 poster_

### Official Review · Reviewer_yWFS · 2024-07-06

**Soundness:** 4
**Presentation:** 4
**Contribution:** 4
**Rating:** 8
**Confidence:** 5

**Summary:**

The paper proposes a method called MambaSCI for efficient reconstruction of quad-Bayer patterned color video snapshot compressive imaging.This method surpasses state-of-the-art algorithms with lower computational and memory costs, providing improved color accuracy and demosaicing.

**Strengths:**

1) The contributions of this paper are quite novel.  The usage of Mamba for video SCI reconstruction has been used for the first time. Moreover, the paper also introduces Quad-Bayer CFA patterned color video SCI reconstruction which is also being done for the first time.
2) The quality of the results produced is quite high. The average PSNR and SSIM scores produced with the method proposed in this paper are both comfortably beating the state of the art results.
3) The paper is well written and the presentation is very clear. The citations, and comparisons with SOTA are sufficient to build confidence in the proposed method.
4) The contributions due to each individual module has been presented which clarifies the necessity for each.

**Weaknesses:**

1) The reconstruction seems to be performed on fairly low resolution images(512 x 512). I would have liked to see the simulations being done on higher resolution.
2) The proposed method is tested only on simulated datasets (since there is no real SCI dataset based on quad-Bayer pattern). To me, it is unclear how the simulated datasets have been created. Was noise added while creating the simulated datasets? My biggest concern is the way the noise has been modeled while creating the simulated dataset might not match the actual data(If there was a way to do it). Therefore, currently the efficacy of the provided solution is somewhat hypothetical.

**Questions:**

1) The paper describes the computational benefits of the proposed approach over SOTA in terms of fewer parameters and FLOPS, however no concrete numbers have been provided around the performance. Is it possible to add some information around the time it would take to reconstruct each frame?
2)  There is no mention on how the masks were generated. Or the conditioning of the masking matrix. This very important information should be included.
3) What is the relationship between T (number of frames used during reconstruction) and the amount of motion between consecutive frames? For instance I would expect you could use more frames(T) if the object was moving slowly within the scene, compared to when it is moving fast. This might help in further improving the computational cost.
4) There is no tiling done during reconstruction. I wonder why this is so? Especially if you want to deal with higher resolution images?

**Limitations:**

Limitations have been adequately addressed in the appendix, so nothing to comment upon here.

---

> ### Author Rebuttal · Authors · 2024-08-07
>
> ### Response to Reviewer yWFS
> Thanks for your valuable comments.
>
> **Q1: Clarification of high resolution images reconstruction.**
>
> **A:** As stated in the main manuscript (lines 233-234), we also evaluated our method on a large-scale simulated dataset. The metrics comparison and visual comparison between our method and the comparison methods for reconstructing 1920x1080 resolution video are presented in Table 4 and Figure 6 in the main manuscript.
>
> **Q2: Noise modeling analysis of simulated datasets**.
>
> **A2**: We have explained how the simulated dataset was created and provided an analysis of our method's effectiveness on a real dataset:
>
> - The middle-scale simulated dataset is created by extracting and cropping frames from high-quality open-source videos available online to obtain the groundtruth (GT), while the large-scale dataset uses videos selected from YouTube as the GT. **Both datasets include noise of real-world scenes.**
> - Given that we are tackling a completely new task, our focus is primarily on addressing the degradation caused by SCI compression and developing effective demosaicing techniques to achieve color fidelity and eliminate artifacts for high-quality reconstruction. The aspect of noise modeling has not yet been fully considered.
> - Through our studies on various Bayer-array-based methods, their performance on simulated and real datasets is positively correlated, thus I believe our method can still achieve significant advantages on real data.
>
> **Q3: Time required to reconstruct each frame.**
>
> **A:** In the table below, we present a comparison of the time required to reconstruct each frame, along with the PSNR and SSIM metrics for the proposed method and other competing methods. Even though our method takes longer than EfficientSCI to reconstruct each frame, it delivers significantly superior performance.
>
> |Method|Time(s)|PSNR(dB)|SSIM
> |-|-|-|-|
> |PnP-FFDnet|0.28|27.86|0.855
> |STFormer|0.20|32.48|0.842
> |EfficientSCI|**0.07**|32.35|0.854
> |MambaSCI|0.16|**35.70**|**0.959**
>
> **Q4: Clarification of how the masks are generated.**
>
> **A:** In the SCI domain, the **mask typically has a specific physical significance**. The basic principle involves modulating different frames within the video cube with varying weights and then integrating the light into the sensor to create a 2D measurement [1-3]. Specifically the masks are the 0-1 matrices that are different from each other in the T-frame. Also this mask is customizable subject to its physical significance and has no effect on the reconstruction results. During training, we simulate the masks of different SCI systems by randomly generating 0-1 matrices to improve the robustness of the system.
>
> **Q5: Clarification of the relationship between T and the amount of motion between consecutive frames.**
>
> **A:** In SCI tasks, $T$ typically represents the compression ratio ($Cr$) and is not directly related to the motion between consecutive frames. A higher $T$ indicates a higher $Cr$ and increased reconstruction difficulty. As highlighted in **Table 1 in the global rebuttal**, both reconstruction difficulty and the required FLOPS escalate with increasing $T$. Nonetheless, our method delivers high-quality reconstruction with superior computational efficiency.
>
> We greatly appreciate the suggestion to use smaller compression ratios for fast motion scenes and larger ones for slow motion scenes to improve efficiency. This insight has inspired us to consider dynamically adjusting the compression ratio based on object movement speed. Thank you for this suggestion, which opens a new avenue for exploration.
>
> **Q6: Clarification of why there is no tiling done during reconstruction.**
>
> **A:**  In the reconstruction process, we first address the SCI compression degradation in the raw domain and then perform demosaicing to convert from the raw domain to RGB. In the raw domain, two common tiling methods involves divideing into non-overlapping patches and splitting the image into four sub-parts: R, G1, G2, and B. We specify why we did not employ either of these methods:
>
> - **Dividing into Non-overlapping Patches.** Dividing an image into non-overlapping patches will lead to localized detail loss. To address this, we employ convolution for shallow feature extraction, which enables us to train effectively on low-resolution images (using a resolution of 128x128 for training and 256x256 for fine-tuning) and process on high resolution images.
> - **Dividing into Four Sub-parts.** This method is efficient for Bayer arrays due to their physical arrangement. However, the quad-Bayer's unique layout poses challenges. Splitting the image into R, G1, G2, and B components requires a more intricate process because each color forms a 2x2 square, which must be considered as a whole rather than isolated pixel by pixel. We attempted this approach, but found it significantly increased both training and inference times.
>
> Thank you for your question, and we will continue exploring efficient tiling methods to achieve lightweight reconstruction for higher-resolution videos.
>
> [1] Yuan X, Liu Y, Suo J, et al. "Plug-and-play algorithms for large-scale snapshot compressive imaging." *CVPR* 2020.
>
> [2] Yuan X, Liu Y, Suo J, et al. "Plug-and-play algorithms for video snapshot compressive imaging." *IEEE TPAMI* 2021.
>
> [3] Yuan X, Brady D J, Katsaggelos A K. "Snapshot compressive imaging: Theory, algorithms, and applications." *IEEE SP* 2021.

---

### Official Review · Reviewer_xnVL · 2024-07-08

**Soundness:** 3
**Presentation:** 3
**Contribution:** 3
**Rating:** 5
**Confidence:** 4

**Summary:**

This paper investigate video snapshot compression imaging reconstruction task by Quad-Bayer CFA pattern into color video SCI. They design a Residual-Mamba-Block consisting of ST-Mamba, Edge-Detail-Reconstruction module and Channel-wise attention module to enhance reconstruction quality and edge details. Experimental results on many datasets validate its effectiveness.

**Strengths:**

It is interesting to introduce Mamba into a new task.

They investigate long sequence modeling to further validate the effectiveness of ST-Mamba.

They proposed a novel framework for video SCI reconstruction task, which outperform existing methods not only in effectiveness but also in efficiency.

**Weaknesses:**

The contribution (ii) and (iii) have limited novelty. It seems that the author simply adapt ST-Mamba to video SCI reconstruction task without elaborated designs. The proposed Residual-Mamba-Block is an incremental combination of ST-Mamba from Vivim, DWConv and Channel-wise Attention. The author should further clarify the novelty.

More description of EDR module should be given. Why EDR module can provide more edge details?

Observed from Table 4, the improvements on large-scale simulation color video are not consistent across datasets. More analysis on such phenomenon should be given. While the author claim the results on Messi and Hummingbird is strained by parameters and FLOPs, specific quantitative comparison should be provided.

**Questions:**

Why we need Mamba instead of self-attention in this task?

Why quad-Bayer pattern is better than a single Bayer-patterned measurement?

**Limitations:**

The author is suggested to compare the efficiency of the proposed method and existing methods on long videos to validate the effectiveness on long sequence.

---

> ### Author Rebuttal · Authors · 2024-08-07
>
> ### Response to Reviewer xnVL
> Thanks for your valuable comments.
>
> **Q1: Clarification of the novelty.**
>
> **A:** We have elaborated on the novelty in lines 160-167 and 289-296 of the manuscript. **Rather than merely adapting ST-Mamba, we introduced the following innovations in video SCI reconstruction:**
>
> - **High Performance Lightweight Network Design:** Although STMamba is an effective lightweight global attention mechanism approach, it still struggles with the high demands of SCI video reconstruction compared to Vivim's medical image segmentation. Our framework enables joint global-local reconstruction, addressing these challenges.
> - **Combined Global-Local Features.** DWConv efficiently extracts local features, avoiding the parameter and complexity issues of common 3D convolutions. **Meanwhile, it is not a straightforward incremental combination with STMamba.** By integrating linear transformations with DWConv features, the network integrates local and global information, enhancing the global features extracted by STMamba. This fusion is crucial for accurately capturing complex edge structures.
> - **Enhanced Channel Interaction:** Many Mamba-based frameworks overlook channel interaction limitations, leading to channel isolation and suboptimal information fusion, which adversely affects reconstruction quality. To address this, we introduce a lightweight channel attention mechanism to improve multi-channel perception and feature representation, refining reconstruction quality.
>
> **Q2: In-depth description of EDR module.**
>
> **A:** We have added a detailed analysis of the EDR module's design principles in lines 184-192, explaining how it enhances video reconstruction by improving the representation of depth features. To better illustrate the EDR module's operation, we redraw the relevant section of Figure 4 in the manuscript. Additionally, further analysis of edge detail reconstruction attributed to the EDR module has been added to the ablation experiment section, specifically in lines 297-299. We additionally provide a visual comparison of the edge detail effect with and without the EDR module on the final reconstruction result (**Figure 3** on pdf).
>
> Specifically, key improvements with EDR include:
>
> - **DWConv :** This technique performs spatial convolution independently on each channel and then applies pointwise (1x1) convolution across channels. It effectively captures localized spatial features, enhancing edge detail perception.
> - **GELU :** Compared to ReLU, GELU handles negative inputs more naturally, improving the model's ability to represent fine details.
> - **Adaptive Weight Initialization:** Truncated normal initialization helps the model capture details more effectively during the early training stages.
> - **Multi-scale Feature Fusion:** Combining linear transformations with DWConv features allows the network to extract global information from local features. This simultaneous processing of global and local information is crucial for understanding complex edge structures.
>
> **Q3: Clarification the reasons for inconsistent improvement effects on large-scale datasets.**
>
> **A:** The inconsistencies may be attributed to:
>
> 1. **Evaluation in Raw Domain:** As described in lines 494-500 of the Supplementary Material, we use additional demosaicing model to show RGB images reconstructed by GAP-TV, PnP-FFDnet, and PnP-FastDvDnet, which **ultimately reconstruct videos in the raw domain**. In contrast, three E2E methods reconstruct RGB video directly. For a fair comparison and excluding the influence of demosaicing, we assess performance in the raw domain. Thus, the RGB video from E2E methods is converted back to the raw domain using a quad-Bayer mask. Figure 6 in manuscript shows that while iterative methods perform better in metrics, they exhibit more visual artifacts.
> 2. **Training Resolution:** The training resolution is 128x128 and fine-tuned at 256x256, which is a gap with large-scale datasets.
> 3. **High-Speed Frame Reconstruction:** The effect of high-speed moving frame reconstruction is to be strengthened under the limitation of the number of parameters and computational complexity. The number of parameters and flops of the model are as follows. Even with the limited FLOPs, we still achieve better performance.
>
> |Method|Params|FLOPs|PSNR|SSIM|
> |-|-|-|-|-|
> |STFormer|**1.23M**|6084.61G|25.11|0.724
> |EfficientSCI|2.21M|11344.367G|24.15|0.710
> |MambaSCI|2.47M|**1957.96G**|**30.80**|**0.896**
>
> **Q4: Clarification of why Mamba and not self-attention.**
>
> **A:** As stated in our response to **Reviewer M9Ao's Q1**, from both the technical perspective, including performance metrics and comparisons of Params and FLOPs as shown in Table 2 of main manuscript, and the perspective of pattern processing, where previous methods based on Transformers and CNNs have struggled with artifacts and color distortions when dealing with quad-Bayer arrays. Mamba has proven to be a more suitable choice for addressing the new challenges we face.
>
> **Q5: Clarification of why quad-Bayer pattern is better than a single Bayer-patterned measurement?**
>
> **A:** As noted in response to **Reviewer M9Ao's Q2**,exploring quad-Bayer-based SCI tasks is both necessary and advantageous. Due to the quad-Bayer's physical structure, where each color is represented by a 2x2 grid, it offers higher resolution, better light intake in low-light conditions, and supports HDR video through different exposure settings.
>
> **Q6: Comparison on long videos.**
>
> **A:** We have conducted new experiments to further validate our method's effectiveness with long sequence videos and also provided corresponding visual comparison images (**Figure 4** on PDF) and a performance comparison table (**Table 1** in global rebuttal). The default $T$ is set to 8, meaning reconstruction is performed after compressing 8 frames of video into 2D observations. To test our method's performance with longer sequences, we also evaluated it with $T = 16$ and $T = 32$.

---

> > ### Comment · Reviewer_xnVL · 2024-08-11
> >
> > 1. About comparison on long videos, when increasing T to 16 and 32, I cannot see the great advantage in run-time and performance over EfficientSCI. Could you please further explain this point?
> > 2. The author claims the global-local design in the proposed framework compared to STMamba. However, as far as I know, there is also Depth-wise Convolution in ST-Mamba. Thus, this cannot be one of the innovations.

---

> ### Author Response · Authors · 2024-08-11
> **Further Explanation**
>
> Thanks for your valuable comments.
>
> **Q1: Clarification about comparison on long videos.**
>
> **A1:** Regarding your questions about run-time and performance, we'll address them as follows:
>
> - **Performance:** As the model's FLOPs increase with $T$, our method maintains only **9.8%** of the FLOPs required by EfficientSCI. While achieving comparable PSNR to EfficientSCI, our method significantly outperforms it in SSIM metrics. Additionally, as shown in the visual comparison in **Figure 4** of the submitted PDF, our method provides superior visualization compared to EfficientSCI.
> - **Run-time:** The reason why our method's runtime is not as competitive as EfficientSCI's can be attributed to two key factors:
>   - **Mamba Characteristics.** Mamba's acceleration is tied to the GPU's hardware characteristics and the underlying CUDA architecture, limiting our ability to enhance its speed directly.
>   - **Network Framework Optimization.** To achieve high-quality reconstruction, our framework currently uses 3D convolution for channel feature interactions at the end of each Residual-Mamba-Block. However, this process becomes more time-intensive as $T$ increases. We plan to optimize the framework further in the future to develop a lightweight, high-quality, and faster reconstruction algorithm.
>
>
> **Q2: Clarification of the innovation of DWConv.**
>
> **A2:** The depth-wise convolution used by Vivim [1] is essentially a 3D depth-wise convolutional layer, which focuses solely on spatial feature extraction. **It lacks a pointwise convolution component, meaning it doesn't account for the relationships between channels**. In contrast, the DWConv we designed **integrates both depthwise and pointwise convolution**. First, it processes spatial information through depthwise convolution, then it combines the relationships between channels via pointwise convolution to produce the final output features. This design allows us to effectively extract local detailed features and combinations between channels while reducing computational costs and the number of parameters.
>
> Thank you for your insightful feedback. We will update the DWConv section of Figure 4 in the main manuscript accordingly and provide a more detailed explanation of DWConv in lines 186-187. Additionally, we will include further analysis in the ablation study in Section 4.5 to enhance the clarity and robustness of our findings.
>
> [1] Yang Y, Xing Z, Zhu L. Vivim: a video vision mamba for medical video object segmentation[J]. arXiv preprint arXiv:2401.14168, 2024.

---

> > ### Comment · Reviewer_xnVL · 2024-08-13
> >
> > Thank you for the rebuttal. Most of my concerns have been addressed. However, I strongly suggest the authors include these additional experiments and analyses in the revised manuscript, and clearly describe the difference against previous Mamba-related works. I tend to maintain my score.

---

### Official Review · Reviewer_zshs · 2024-07-13

**Soundness:** 2
**Presentation:** 3
**Contribution:** 1
**Rating:** 3
**Confidence:** 5

**Summary:**

This manuscript introduces the Mamba model and Quad-Bayer CFA pattern into color video snapshot compressive imaging (SCI) for the first time. Specifically, the proposed MambaSCI adopts a non-symmetric U-shaped encoder-decoder architecture, which includes DWConv, Residual-Mamba-Blocks, and ReConv. The Residual-Mamba-Blocks integrate several modules designed to enhance reconstruction quality and edge details. Experiments demonstrate that MambaSCI outperforms comparison methods in both quantitative and qualitative results, with fewer parameters and FLOPs.

**Strengths:**

1.	This work introduces a method for quad-Bayer patterned color video SCI for the first time. Quad-Bayer sensors offer better hardware performance than Bayer pattern sensors and represent a promising direction worth exploring in SCI.
2.	The Mamba model, as a popular module for extracting data causality, aligns well with SCI tasks. Its incorporation into SCI is well-justified.
3.	The manuscript is generally well-structured and well-written.

**Weaknesses:**

+ The proposed MambaSCI takes X_in as input, which is obtained from initialization Block. So the framework can be seen as a simple video enhancement task. Furthermore, MambaSCI does not consider the characteristics of Quad-Bayer pattern like previous work (STFormer, EfficientSCI), but instead directly adopts or integrates popular modules.The in-depth analysis appears to be lacking in certain modules, specifically within the passages from lines 185-187 and 193-197.

+The lack of an explanation for the initialization block raises questions about whether special processing was done to address issues introduced by the quad-Bayer pattern. It would be better to add further visualization of the X_in results for comparison.

+In Figure 6, the results of proposed method show artifacts and color distortion in the lower left corner of ``Swinger”. More analysis should be included.

+The manuscript lacks real experiments on videos captured by quad-Bayer patterns. All the results are obtained on  simulated data. Results on real data are expected to evaluate the effectiveness of the proposed method.

**Questions:**

Please refer to weakness.

**Limitations:**

Please refer to weakness.

---

> ### Author Rebuttal · Authors · 2024-08-07
>
> ### Response to Reviewer zshs
> Thanks for your valuable comments.
>
> **Q1: Differences between the proposed method and the video enhancement-based ones.**
>
> **A:** **MambaSCI significantly differs from video enhancement technology in the nature of the task, input differences, and use of prior knowledge**, as detailed below:
>
> - **Nature of the Task:** Video enhancement tasks typically improve RGB video quality by addressing noise or missing frames. In contrast, the MambaSCI network addresses the more complex SCI task, which involves not only overcoming compression degradation but also handling the additional effects of the quad-Bayer array mask. Thus, the MambaSCI network has a dual mission: **accurate reconstruction of compressed data** and **conversion and demosaicing from the raw domain to standard RGB format**, requiring highly flexible and capable of managing the unique complexities of SCI technology.
> - **Huge Gap Between the Input of Two Tasks:** Same to previous models like STFormer and EfficientSCI, our MambaSCI reconstruction network also requires an initialization block for up-sampling, which is a crucial component of the SCI reconstruction task. However, the $\mathbf{x_{in}}$ obtained after initialization differs significantly from traditional video enhancement inputs. We present $\mathbf{x_{in}}$ after initialization in **Figure 1** in supplemental PDF, where it is evident that the degradation is much more severe than what common video enhancement networks can handle.
> - **Utilization of Different Prior Knowledge:** In the reconstruction process, MambaSCI leverages the unique priori knowledge of compression and quad-Bayer arrays, contrasting sharply with traditional video enhancement’s traditional focus on smoothness and edge preservation.
>
>
> **Q2: Clarification of the quad-Bayer pattern is not characterized in the same way as in previous work.**
>
> **A:** Both STFormer and EfficientSCI were designed as Bayer array-based video compression reconstruction methods, only considered Bayer pattern features in their initialization blocks. **In contrast, we have developed a specialized initialization block to address the unique characteristics of quad-Bayer physical arrays, and applied it specifically to GAP-TV, PnP-FFDnet, and PnP-FastDVDnet.** Details are as follows:
>
> - GAP-TV, PnP-FFDnet, and PnP-FastDVDnet use iterative optimization by dividing the raw domain into R, G1, G2, and B blocks for independent reconstruction. Similarly, we divided the quad-Bayer array into R, G1, G2, and B sub-blocks as shown in lines 103-107 in the main manuscript. However, due to the quad-Bayer pattern's 2x2 grid representation for each color, processing is significantly more complex than standard Bayer pattern, resulting in an increased processing time from 0.0015s to 7.3312s at 512x512 resolution compared to the SCI generic initialization method.
> - Leveraging the powerful modeling capabilities of the E2E model and considering the need for a lightweight design due to the limited computing power of future handsets, we selected the SCI generic initialization block for STFormer, EfficientSCI, and MambaSCI. Our experiments confirmed nearly no performance loss with this choice.
>
> **Figure 1** of our PDF submission provides a visual comparison of $\mathbf{x_{in}}$ obtained using the SCI generic initialization block versus an initialization block designed specifically for the physical characteristics of a quad-Bayer array.
>
> **Q3: Enhancing in-depth analysis of certain modules.**
>
> **A:** We have added a deeper analysis of the EDR module, bottleneck layer, and the decoding layer in the following areas:
>
> - **More Detailed Analysis:** We have thoroughly analyzed the EDR module, detailing its inner workings and its critical role in our framework. Additionally, we have improved the presentation and analysis of the bottleneck and decoding layers.
> - **Figure-Text Interaction:** We have redrawn Figures 3 and 4 in the main manuscript and improved their integration with the text to facilitate better understanding and clarity for the reader.
> - **Corresponding to the Ablation Experiment:** The module description is linked to the associated ablation experiments, allowing the reader to intuitively grasp the module's important role in the network.
>
> **Q4:  Lack of real experiments on videos captured by quad-Bayer patterns.**
>
> **A:** The reasons for the lack of real experiments are as follows:
>
> - Since we are exploring a new task and a new discovery that is not currently being investigated, there are no publicly available datasets for comparison, as stated by reviewer **yWFS (there is no real SCI dataset based on quad-Bayer pattern)**.
> - To obtain the real-world datasets, a corresponding optical coding system must be assembled. We are working on building this system, but it will take six months to one year to complete, even with experienced personnel. Additionally, we need to upgrade the existing system by replacing the current modules with four-layer modules.
> - To demonstrate the advantages of quad-Bayer over Bayer, we plan to acquire more datasets from extreme scenes and collect HDR data. This will significantly expand the current research field of SCI.
>
> **Q5: Analysis of artifacts and color distortion in the lower left corner of "Swinger".**
>
> **A:** Thanks to your careful observation, we have added the visual comparison in **Figure 2** of the submitted PDF, showing different methods to locally zoom in on the lower left corner of the "swinger." The intricate ropes in this area, as depicted in the GT, present a significant challenge due to their dense packing and rapid movement between frames, leading to ghosting artifacts in the reconstruction.
>
> To address this limitation, we plan to refine our model further. In future work, we will focus on improving the model's ability to handle fast motion between frames, minimizing artifacts and achieving more accurate reconstructions in complex and dynamic scenes.

---

> > ### Comment · Reviewer_zshs · 2024-08-08
> >
> > 1. The quad-Bayer pattern has some advantages, e.g., higher resolution and HDR capability, in convential camera imaging. However, these advantages are not necessarily inherited by the SCI configuration due to the mask modulation. Moreover, the current simulation of SCI pipline is oversimplified without considering optical effects in real scenarios. Therefore, it is crucial to have real-scene evidence to verify the significance of the SCI  task with quad-Bayer.
> > The manuscript lacks real experiments on videos captured by quad-Bayer patterns. All the results are obtained on simulated data. Results on real data are expected to evaluate the effectiveness of the proposed method.
> >
> > 2. Regarding the result of the "Swinger", I feel that the PnP-FastDvDnet method provides better  visual quality than the proposed MambaSCI, given that both introduced some artifacts. While PnP-FastDvDnet presents some ghosting artifacts, MambaSCI brings obvious geometrical distortion: line structures are curved locally with also some ringing artifacts.

---

> ### Author Response · Authors · 2024-08-10
> **Further explanation**
>
> Thanks for your valuable comments.
>
> **Q1: Clarification on the lack of real experiments using videos captured by quad-Bayer patterns.**
>
> **A1:** As we are pioneering a **new task** in the field of video SCI, which is still in its **exploratory stage**, no public dataset currently exists for this class. Meanwhile, our main innovations include:
>
> - **New task:** Quad-Bayer offers significant advantages over traditional Bayer sensors, such as higher resolution and HDR capabilities, and is widely used in smartphone cameras. However, there have been no studies on its application in video SCI. We are **the first** to introduce quad-Bayer into video SCI, aiming to expand the field and achieve higher quality reconstruction.
> - **New method:** To enable lightweight reconstruction, we are **the first** to utilize an asymmetric U-shaped architecture, employing customized Residual-Mamba-Blocks modules to achieve efficient, high-quality video reconstruction.
>
> Furthermore, constructing the corresponding SCI encoding system requires specialized optical encoding hardware and significant time, which is often beyond the capabilities of researchers focused on decoding algorithms.
>
> **We are working on building and refining this system** to fully leverage the advantages of quad-Bayer over Bayer, such as capturing video in low-light environments and compressing HDR video SCI by adjusting pixel exposure within the 2x2 color grid.
>
> **Q2: Clarification of swinger's visual comparison.**
>
> **A2:** As detailed in lines 494-500 of the Supplementary Material of main manuscript, we utilized the high-performance BJDD [1] model for joint denoising and demosaicing to display the RGB images reconstructed by PnP-FastDvDnet. Specifically, **PnP-FastDvDnet only handles Raw domain reconstruction**, while BJDD performs further denoising and demosaicing to obtain the final RGB video. In contrast, **our model integrates both Raw domain reconstruction and demosaicing**. As a result, the visual quality of PnP-FastDvDnet may appear superior due to the BJDD network's influence. Therefore, as described in the main text, to ensure a fair comparison, we converted the RGB video output from MambaSCI back to the Raw domain and calculated metrics such as PSNR and SSIM in the Raw domain. The performance metrics of MambaSCI vs. PnP-FastDvDnet on swinger in Raw domain are compared as follows:
>
> |Methods|PSNR (dB)|SSIM|
> |-|-|-|
> |PnP-FastDvDNet|28.60|0.887|
> |MambaSCI|**29.78**|**0.920**|
>
> As seen in the table, our method outperforms PnP-FastDVDnet in both PSNR and SSIM in the Raw domain. Additionally, as shown in Figure 6 and Figure 15 of the main manuscript, our approach achieves superior reconstruction in other detailed areas compared to PnP-FastDVDnet.
>
> [1] A Sharif S M, Naqvi R A, Biswas M. "Beyond joint demosaicking and denoising: An image processing pipeline for a pixel-bin image sensor." *CVPR* 2021.

---

> > ### Comment · Reviewer_zshs · 2024-08-11
> > **Further discussion**
> >
> > I appreciate the authors' clarification and the new result in the Raw domain. However, I am not sure whether my concerns have been addressed.
> >
> > **Q1: Real System.**
> > For simulation as is, changing the Bayer pattern is a simple modofication by easily replacing the old Bayer mask by the new mask in the data generation and reconstruction. Moreover, the simulation is oversimplified without considering the camera response, optical transmission, and mask discretization ($\mathbf{M} \in \mathbb{R}$ in current work), which present significant gap against practical implementation.  Glad to know that the authors are constructing a real system for the proposed pipeline, which I feel that would significantly lift the contribution and quality of this research work.
> >
> > **Q2: Reconstruction Quality.**
> > Intuitively, if trained properly, the proposed end2end reconstruction pipeline has a great potential to outperform the cascaded “PnP-FastDvDnet + BJDD” combination. If it is not the case, I feel that it would be worth to further improve the reconstruction net.

---

> ### Author Response · Authors · 2024-08-11
> **Further Explanation**
>
> Thanks for your valuable comments.
>
> **Q1: Real System.**
>
> **A1:** Since we are exploring a new task in the SCI field, our focus has been on addressing issues like artifacts and color  distortion when dealing with quad-Bayer array video. We appreciate your insight into the potential challenges that might arise in real-world scenarios. We will take your suggestions into account and work on developing a quad-Bayer-based coding system as soon as possible.
>
> **Q2: Reconstruction Quality.**
>
> **A2:** Our proposed E2E network strikes a balance between performance and efficiency. As demonstrated in **Figure 5 of the main manuscript**, at a resolution of 512x512, our method produces superior visual results compared to the cascaded "PnP-FastDvDnet + BJDD" combination. Our approach maintains high color fidelity without introducing additional artifacts, whereas the "PnP-FastDvDnet + BJDD" combination suffers from severe artifacts and shape distortions across multiple datasets. Also, we provide a comparison of the corresponding performance metrics as well as a comparison of the reconstruction times in the table below:
>
> *Table 1: A comparison of performance metrics (PSNR (dB), SSIM) and reconstruction time between PnP-FastDVDnet and MambaSCI.* (**Note: The time for PnP-FastDvDnet does not include the additional time required for the demosaicing process performed by BJDD.**)
>
> |Method|Beauty|Bosphours|Runner|ShakeNDry|Traffic|Jockey|Running time(s)|
> |-|-|-|-|-|-|-|-|
> |PnP-FastDvDnet|34.29,0.967|33.07,0.947|34.18,0.928|30.11,0.883|23.74,0.811|32.70,0.921|14.60|
> |MambaSCI|**36.95,0.979**|**38.62,0.982**|**40.02,0.977**|**34.55,0.950**|**27.52,0.904**|**36.54,0.960**|**5.12**|
>
> Table 1 clearly highlights the superior performance of our approach compared to PnP-FastDVDnet, along with a significant advantage in the time required for reconstruction.
>
> Additionally, we are actively exploring improvements to the pipeline and have identified the following areas for enhancement:
>
> - **Raw Domain Reconstruction Performance:** While our method performs well on middle-scale datasets, it does not achieve superior metrics on large-scale datasets, primarily due to the significant difference in resolution sizes between training (128\*128, and 256\*256 for finetuning) and testing (1920\*1080). We are trying to further investigate multi-scale generalization approaches to address this issue.
> - **Improved Demosaicing Algorithms:** Similar to previous Bayer-based reconstruction methods, and for lightweight considerations, our current approach utilizes 3D convolution for the final demosaicing operation. However, the distinct structure of the quad-Bayer pattern makes it more challenging to demosaic compared to traditional Bayer arrays. This complexity can often result in color confusion and artifacts during the demosaicing process. **Recognizing this limitation, we have designed a lightweight demosaicing network that has already shown promising results in preliminary experiments.** We will continue to refine and explore this approach further.

---

> > ### Author Response · Authors · 2024-08-13
> >
> > We sincerely appreciate your careful consideration and prompt response to our rebuttal. We are readily prepared to address any further questions or concerns you may have.

---

> > ### Comment · Reviewer_zshs · 2024-08-13
> >
> > Thanks for the explanation, from which I can feel their frank and honest sense. While this work investigate a topic that might be interested in the area of SCI, it leaves too many aspects unconsidered, such as camera response, optical transmission, noise modelinng (Commented also by Reviewer yWFS), and mask discretization. This suggests that the research is at its early stage, and I would encourage the authors further investigate these issues.

---

### Official Review · Reviewer_M9Ao · 2024-07-15

**Soundness:** 3
**Presentation:** 3
**Contribution:** 3
**Rating:** 6
**Confidence:** 5

**Summary:**

This paper presents a method for compressive video image using Mamba-Unet for quad bayer sensors.

**Strengths:**

+ The work seems to be in a less explored area of research.

**Weaknesses:**

- The work is not making significant contribution in terms of method. Directly applying Mamba-Unet to this problem looks unnatural. Why not use a transformer or a CNN? I dont think there will be a much of a difference.

- The writing needs improvement and section 3 is not that clear.

**Questions:**

What is the primary motivation to address quad bayer sensors when they are not that prevalent?

What is the need of using Mamba-Unet for this work?

**Limitations:**

Limitations are not addressed well.

---

> ### Author Rebuttal · Authors · 2024-08-07
>
> ### **Response to Reviewer M9Ao**
> Thank for your valuable comments.
>
> **Q1. Clarification of why Mamba and not transformers or CNNs.**
>
> **A:** We have added further discussion of this topic in lines 59-61 of main manuscript, with a more detailed analysis and experimental validation as follows:
>
> Transformer and CNN-based Video Snapshot Compressive Imaging methods have been proposed such as **transformer-based method STFormer** and the **hybrid CNN- and transformer-based method EfficientSCI**. However, existing methods still face two issues:
>
> 1. **Technique perspective:**
>    - The high computational complexity of Transformers and the lack of a global attention mechanism in CNNs hinder their extension into modern, lightweight, and efficient network architectures. Therefore, we focus on exploring multi-scale reconstruction of Mamba-Unet into the Quad-Bayer pattern to achieve lightweight design and enable deployment on mobile devices.
>    - As shown in Table 2 of main manuscript, MambaSCI-B outperforms STFormer and EfficientSCI, using only **31%** of STFormer’s parameters and **4.5%** of its FLOPS, and **69%** of EfficientSCI's parameters and **9.8%** of its FLOPS.
> 2. **Pattern perspective:** All existing video SCI methods are designed based on the traditional Bayer pattern. When applied to videos captured by quad-Bayer cameras, these methods often result in color distortion and ineffective demosaicing, rendering them impractical for primary equipment. Thus, we aim to solve these issues with two key contributions: **new task and new method**. That is, efficient and qualitative recovery of quad-Bayer arrays through the asymmetric Mamba-Unet framework. **To the best of our knowledge, we are the first one to develop the Quad-Bayer Patterned Video Snapshot Compressive Imaging task and the first algorithm to elaborate designs of Mamba-UNet to this task.**
>
> **Q2.  Clarification of the necessity and advantages for quad-Bayer .**
>
> **A:** Quad-Bayer arrays are prevalent and almost all current flagship smartphones cameras, such as the iPhone 14 Pro/Max, vivo X90 Pro+, Xiaomi 13S Ultra, and OPPO Find X6 Pro, utilize quad-Bayer arrays. We have added a description of the main applications where quad-Bayer is used and its advantages over Bayer in lines 35-38 of the main manuscript. Meanwhile, integrating quad-Bayer array into video SCI not only is necessary but also offers significant advantages over Bayer arrays:
>
> 1. **Necessity:**
>    - **Widespread Use on Smartphones:** Quad-Bayer arrays are common in smartphone cameras [1], which are frequently used for video recording. Existing methods face color distortion and artifacts, which our work aims to provide a customized solution for Quad-Bayer color video SCI tasks to overcome these issues while offering space-saving, high-quality reconstruction through compression and lightweight construction on smartphones.
>    - **New Industrial Demands and Research Trends:** Bayer arrays have been widely used in industry and well-studied in academia due to their long history, performing adequately in bright scenes. However, the imaging capabilities in low-light conditions are limited. Therefore, exploring quad-Bayer arrays is essential to address these deficiencies.
> 2. **Advantage:** Quad-Bayer arrays has the following advantages over Bayer arrays:
>    - **Higher Resolution:** Quad-Bayer provides higher resolution than Bayer arrays by realizing higher pixel density on the sensor for better detail reproduction compared to conventional Bayer arrays [2].
>    - **Low-Light Performance Enhancemente:** Quad-Bayer collects more light in low-light environments, improving the sensor's signal-to-noise ratio and reducing noise interference.
>    - **HDR Capability:** By adjusting quad-Bayer sub-pixels and setting different exposure values on different sub-pixels, high dynamic ranging (HDR) images or videos can be captured to improve the detail performance of highlights and shadows [3-4].
>    - **Color Accuracy:** Quad-Bayer pattern captures richer color information by sampling red, green and blue colors in four directions. This fine sampling reduces the likelihood of color distortion and improves the color accuracy and realism of the image [5].
>
> **Therefore, we have introduced quad-Bayer arrays to video SCI for the first time, providing a solution for reconstructing higher-quality videos even HDR videos in the future.**
>
> **Q3. Section 3 lacks clarity.**
>
> **A:** To address this, we have made the following changes to the paper:
>
> - **Top-Down Structure:** We use a top-down structure to introduce our framework, providing an overview followed by detailed component breakdowns. This approach ensures a clear understanding of the overall algorithm and the specific functions of each individual part.
> - **Clear Delineation:** We clearly delineate Section 3.2 to provide a detailed analysis of each module, including a thorough examination of the internal structure and a clear explanation of each module's function.
> - **Figure-Text Interaction:** Figures 3 and 4 in the main manuscript have been revised to align with the content and more effectively present the network framework and internal modules details.
> - **Contextual Analysis:** We ensured the model description is consistent with Section 4’s experimental setup and interacts with the Supplementary Material’s pseudo-code to aid reproduction.
>
> [1] Madhusudana P C, et al. "Mobile Aware Denoiser Network (MADNet) for Quad Bayer Images" *CVPRW* 2024.
>
> [2] Zheng B, et al. "Quad Bayer Joint Demosaicing and Denoising Based on Dual Encoder Network with Joint Residual Learning" *AAAI* 2024.
>
> [3] Kim J, Kim M H. "Joint demosaicing and deghosting of time-varying exposures for single-shot hdr imaging." *ICCV* 2023.
>
> [4] Wu T, et al. "High Dynamic Range Imaging with Multi-Exposure Binning on Quad Bayer Color Filter Array. " *ICIP* 2023.
>
> [5] Lee H, et al. "Efficient unified demosaicing for bayer and non-bayer patterned image sensors." *ICCV* 2023.

---

> > ### Comment · Reviewer_M9Ao · 2024-08-10
> >
> > Thanks for the detailed response. My doubts are cleared and such a revision should help. I have increased my rating after also reading other reviews. Good luck!

---

> ### Author Response · Authors · 2024-08-11
> **Thank you for your review！**
>
> We sincerely appreciate your thorough review of our paper, careful consideration of our rebuttals, and raising the score. If you have any further questions or concerns, we are always available to provide additional clarification as necessary.

---

### Author Rebuttal · Authors · 2024-08-07

We thank all the reviewers for their time and insightful comments which have helped improve our paper. We are pleased that the reviewers found our introduction of Mamba into the SCI task to be very novel and well justified, and that our proposed quad-Bayer patterned SCI task is a direction worth exploring. Some of the main criticisms are summarized in the next section, after which we will address the comments of the delicious reviewers separately.

In response to the reviewers' constructive criticism, we added some new experiments including differential analysis of whether considering quad-Bayer feature in the initialization block, analysis of Swinger data results, evaluation of the EDR module for edge detail reconstruction, and a comparison of reconstruction in higher frame rate scenes. The new analyses are shown in **Figures 1 to Figure 4** in the associated PDF file.

**New comparative experiments and analysis.** To verify the validity and reasonableness of our method, we have conducted additional experiments and analysis accordingly. These experiments include:

- **Comparison of Initialization Methods with Quad-Bayer Characteristics and SCI Generic Blocks.** We have customized a Quad-Bayer-specific initialization block by considering the characteristics of the Quad-Bayer pattern (each color consists of 2x2 pixels) and have applied it to GAP-TV, PnP-FFDnet, and PnP-FastDVDnet. **Figure 1** shows a visual comparison of the $\mathbf{x_{in}}$ obtained considering Quad-Bayer characteristics in the initialization block and the $\mathbf{x_{in}}$ obtained using SCI generic initialization block, demonstrating the use of the SCI generic initialization block can reduce reconstruction time while maintaining performance. For a more detailed analysis, see **Reviewer zshs's Q2**.
- **Zoomed-In Patch Comparison of "Swinger" with Ghosting Artifact Analysis.** **Figure 2** shows a localized zoomed-in patch comparison image of the lower left corner of the "swinger", along with an analysis of the ghosting artifacts that appear.
- **Impact of EDR Module on Edge Detail Reconstruction.**  **Figure 3** shows the effect of w/ or w/o EDR module on the impact of edge detail reconstruction, we can see the EDR module can significantly improve the reconstruction of edge details.
- **Reconstruction Effects at Higher Frames and Compression Ratios.** We conducted new experiments in longer frame rates and larger compression rate scenarios. **Figure 4** shows a comparison of the reconstruction effects of our method and the comparative methods at higher frames T (compression ratios). The related performance comparison is shown in **Table 1**:

*Tabel 1: Performance analysis at $T$=16 and 32 cases. Our MambaSCI outperforms in PSNR and SSIM while requiring less than **10%** of the FLOPS of the comparison method.*
| T  | Methods     | Params (M) | FLOPS (G)  | PSNR (dB) | SSIM   | Time (s) |
|----|---------------|-----------|-------|---------|-----------|----------|
| 16 | PnP-FFDnet    | -       | -        | 24.85     | 0.767   | 4.52     |
|    | STFormer      | 19.49   | 24311.76 | 25.21    | 0.685    | 3.13     |
|    | EfficientSCI  | 8.83    | 11406.23| 25.35    | 0.656    | **1.37**     |
|    | MambaSCI      | **6.11**    | **1113.78**  | **25.39**    | **0.817**   | 2.67     |
|
| 32 | PnP-FFDnet    | -       | -         | 1.82     | 0.496 | 9.85     |
|    | STFormer      | 19.49     | OOM      | -     | -  | -    |
|    | EfficientSCI  | 8.83     | 22825.34      | **23.24**    | 0.653  | **2.21**    |
|    | MambaSCI      | **6.11**     | **2227.57**      | 22.44    | **0.785**  | 5.20     |

---

### Author Response · Authors · 2024-08-14
**Summarization**

As the discussion phase comes to a close, we would like to summarize our key points for the convenience of the esteemed reviewers and ACs.

**New Task:**  As far as I know, we are the first to introduce quad-Bayer arrays into the field of video SCI, which responds to the trend of smartphones becoming the mainstream video shooting devices and the mainstreaming of quad-Bayer arrays in smartphone cameras, and also compensates for the serious color distortion and ghosting problems that exist in the existing methods when facing quad-Bayer array videos. Meanwhile, numerous literatures have shown that quad-Bayer arrays can provide better image quality compared with Bayer, and we hope to broaden the research scope and reconstruction quality of video SCI.

**New Approach:** For lightweight video reconstruction, we have pioneered the use of an asymmetric U-shaped architecture with customized Residual-Mamba-Blocks modules. This design enables efficient and high-quality video reconstruction, achieving superior results with fewer parameters and FLOPS compared to previous transformer and CNN-based methods, as demonstrated in **Table 2** of the main manuscript.

**Real Dataset:** As the task is still in the exploratory phase, public datasets of this class are currently targeted. Additionally, constructing the necessary SCI encoding system requires specialized optical hardware and considerable time, which often exceeds the resources of researchers focused on decoding algorithms. We are committed to developing and refining this system to fully capitalize on the advantages of quad-Bayer over Bayer. We also extend our gratitude to reviewers **zshs** and **yWFS** for highlighting potential challenges in real-world applications, which will guide us in building a robust system.

Finally, we sincerely thank all reviewers for their time and thoughtful reviews. We will carefully consider your constructive suggestions and continue to improve our work.

---

### Decision · Program_Chairs · 2024-09-25

**Decision:**

Accept (poster)

**Comment:**

This paper received 4 mixed reviews with the scores spread across a spectrum - one 3 (Reject), one 5 (Borderline Accept), one 6 (Weak Accept), and one 8 (Strong Accept).

There was a general appreciation for the addressed problem (video compressive imaging with quad Bayer sensors, which is timely given the increasing popularity of quad-Bayer sensors), quality of the exposition, the principled nature of the proposed solution which is well-motivated, and the overall quality of the results.

There were several concerns raised, in particular about lack of real experiments with a quad Bayer sensor. The authors submitted a response and there was considerable discussion post-rebuttal. While the concern around lack of real experiments remain (due to the relative difficulty of obtaining large scale quad-Bayer datasets), most other concerns were reasonably addressed. Therefore, on balance, an accept decision is recommended.

The authors are strongly urged to incorporate the reviewers' feedback and include the additional results / evaluations promised in the rebuttal when preparing the final camera-ready version.